# Sensitive dissection of a genomic regulatory landscape using bulk and targeted single-cell activation

## Graphical abstract

## Authors

Dubravka Vučićević, Che-Wei Hsu,
Lorena Sofia Lopez Zepeda, ...,
Markus Landthaler, Scott Allen Lacadie,
Uwe Ohler

## Correspondence

vucicevic.dubravka@gmail.com (D.V.),
scott.lacadie@mdc-berlin.de (S.A.L.),
uwe.ohler@mdc-berlin.de (U.O.)

## In brief

Vučićević et al. describe TESLA-seq, which combines pooled CRISPR activation with targeted single-cell RNA-seq to map enhancer-gene connections at high sensitivity. TESLA-seq was applied across a genomic region surrounding the PHOX2B gene, a key transcription factor affecting the growth of neuroblastoma cells, to reveal 60 regulators for 30 genes.

## Highlights

- CRISPRa tiling screen reveals hundreds of growth-modulating elements surrounding PHOX2B

- TESLA-seq finds gene targets of dozens of candidate CRISPRa-responsive elements (CaREs)

- Validated regulatory CaRE-gene interactions and profiled their genomic features

- Integrated epigenomics data identify tissues where CaRE-gene pairs are active

Vučićević et al., 2025, Cell Genomics 5, 100984
October 8, 2025 © 2025 The Author(s). Published by Elsevier Inc.

# Cell Genomics

CellPress

## Article

# Sensitive dissection of a genomic regulatory landscape using bulk and targeted single-cell activation

Dubravka Vučićević,[1,*] Che-Wei Hsu,[1,5,6] Lorena Sofia Lopez Zepeda,[1,4] Martin Burkert,[1,4] Antje Hirsekorn,[1] Ilija Bilić,[1,7] Nicolai Kastelić,[2] Markus Landthaler,[2] Scott Allen Lacadie,[1,*] and Uwe Ohler[1,3,8,*]

[1]Computational Regulatory Genomics, Berlin Institute for Medical Systems Biology of the Max Delbrück Center for Molecular Medicine in the Helmholtz Association, 10115 Berlin, Germany
[2]RNA Biology and Posttranscriptional Regulation, Berlin Institute for Medical Systems Biology (BIMSB), Max Delbrück Center for Molecular Medicine, 10115 Berlin, Germany
[3]Department of Biology, Humboldt Universität Berlin, 10117 Berlin, Germany
[4]These authors contributed equally
[5]Present address: Department of Biology, Duke University, Durham, NC 27708, USA
[6]Present address: Howard Hughes Medical Institute, Duke University, Durham, NC 27708, USA
[7]Present address: BioNTech Cell & Gene Therapies GmbH, 55131 Mainz, Germany
[8]Lead contact
*Correspondence: vucicevic.dubravka@gmail.com (D.V.), scott.lacadie@mdc-berlin.de (S.A.L.), uwe.ohler@mdc-berlin.de (U.O.)

## SUMMARY

Enhancers are known to spatiotemporally regulate gene transcription, yet the identification of enhancers and their target genes is often indirect, low resolution, and/or assumptive. To identify and functionally perturb enhancers at their endogenous sites, we performed a pooled tiling CRISPR activation (CRISPRa) screen surrounding PHOX2B, a master regulator of neuronal cell fate and a key player in neuroblastoma, and found many CRISPRa-responsive elements (CaREs) that alter cellular growth. To determine CaRE target genes, we developed TESLA-seq (targeted single-cell activation), which combines CRISPRa screening with targeted single-cell RNA sequencing and enables the parallel readout of the effect of hundreds of enhancers on all genes in the locus. While most TESLA-revealed CaRE-gene relationships involved neuroblastoma-related regulatory elements, we found many CaREs and target connections normally active only in other tissues. This highlights the power of TESLA-seq to reveal gene regulatory networks, including edges active outside of a given experimental system.

## INTRODUCTION

One of the critical questions in molecular biology is how gene expression is regulated in a temporal and tissue-specific manner in both health and disease. The genome-wide mapping of open chromatin and histone modifications indicative of transcriptional regulatory mechanisms and their states has provided large compendia of cis-regulatory elements (CREs). However, identifying the target genes of regulatory elements (REs) is challenging: CREs do not necessarily regulate their closest gene, may regulate several genes, and may only show a functional effect in combination with other CREs. Approaches to determining CRE-target relationships include measurements of 3D proximity between CREs and candidate targets, CRE-gene co-activity correlations across many cell types or states, and CRE perturbation with transcriptional activity readouts.[1–5] The first two have the potential advantage of being high throughput and the disadvantage of being purely correlative without direct functional evidence, whereas traditional perturbation approaches provide

such direct evidence but are comparatively low throughput. Despite numerous computational and experimental methods for predicting genetic cis-regulatory elements (CREs) based on sequence and chromatin features, identifying functional CREs and their targets has remained challenging.[2–6]

For a long time, the functional characterization of CREs, such as enhancers, has been performed outside their genomic context using reporter assays.[1–5,7] The development of CRISPR-Cas9 technologies now allows the examination of CREs at their native locus. Specifically, engineered fusion proteins can endogenously activate (CRISPR activation [CRISPRa]) or inhibit (CRISPR inhibition [CRISPRi]) CREs, and we can perform large-scale assays to study thousands of CREs in a single experiment using pooled CRISPR-Cas9 screens.[8,9]

Transcriptome-wide measurements of RNA in up to hundreds of thousands of single cells in a single study have revolutionized cell-type quantification from heterogeneous samples. Single-cell CRISPR screening approaches hold great promise for overcoming the low-throughput disadvantage of CRE perturbation

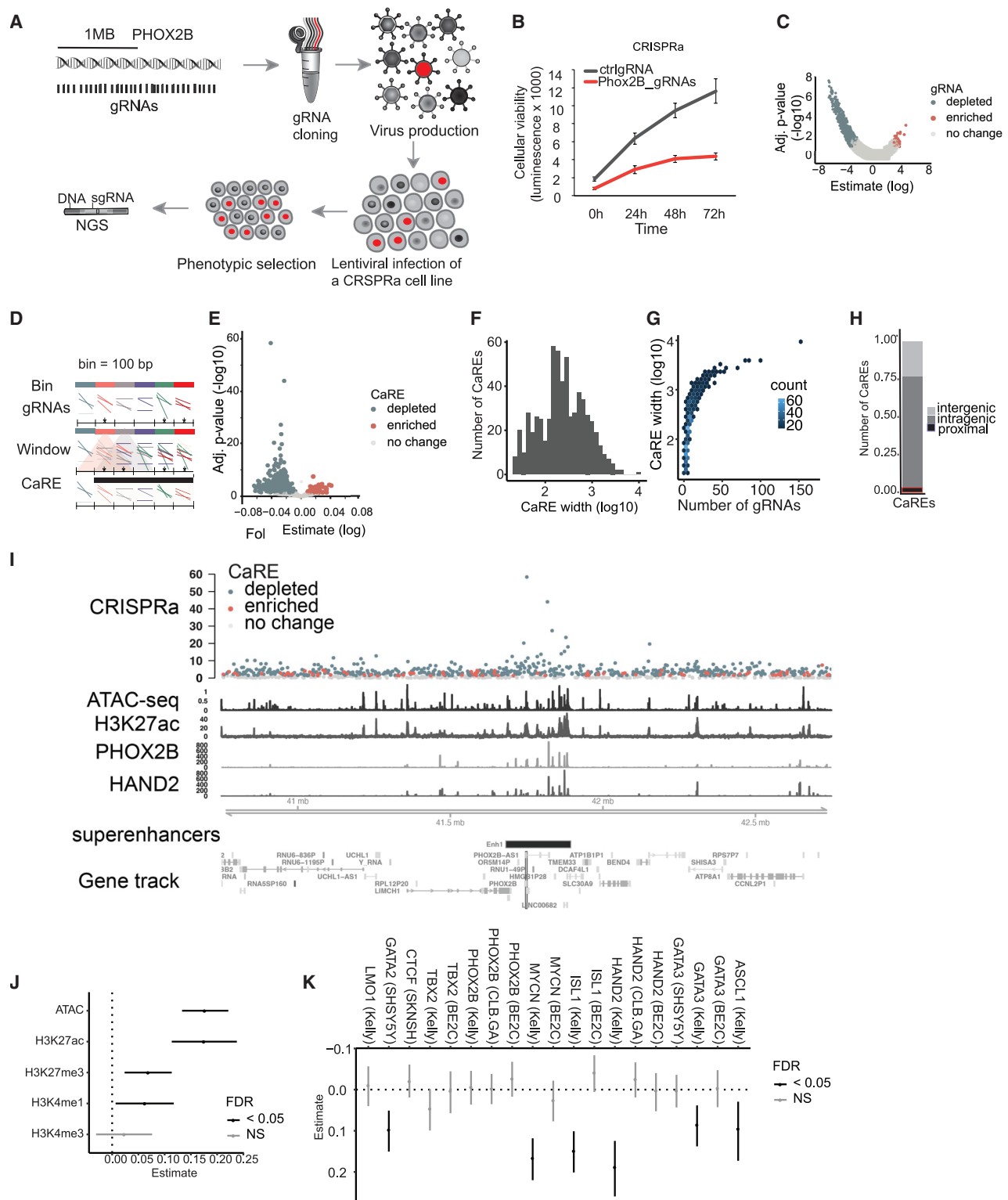

**Figure 1. Bulk phenotypic CRISPR activation screen reveals genetic regulatory elements in the PHOX2B genomic landscape**

(A) Overview of the CRISPRa screen.

(B) MTT viability assay with control or gRNAs targeting the promoter of PHOX2B in an SH-SY5Y-VPR (CRISPRa) cell line.

experiments,[10–18] yet there are some limitations to these approaches. First, single-cell RNA sequencing (scRNA-seq) screens have so far largely employed wild-type Cas9 and the dCas9-KRAB transcriptional repressor construct to perturb enhancers.[10–18] Although they can provide valuable insight into enhancer biology, these Cas9 constructs are less effective at perturbing distal CREs than they are for proximal CREs.[19–21] Second, selecting CREs based on pre-existing features, such as chromatin accessibility and the presence of histone marks,[14] introduces potential biases from the quality and completeness of such annotations and our interpretation of them. Third, scRNA-seq approaches are costly and extremely sparse, leading to unreliable assessment of differential gene expression, especially for lowly expressed genes.[10–14,18] Single-cell perturbations of CREs do not require full transcriptome quantification due to known spatial restrictions between CREs and their targets and, therefore, benefit greatly from targeted sequencing via the substantial reduction of dropout rates for genes of interest.[10–17]

To overcome current limitations, we propose a two-step strategy for identifying REs. In the first step, we assay tens of thousands of potential REs in a tiling CRISPRa screen for their effect on the cellular phenotype. In the second step, we re-examine hundreds of these elements that affected the phenotype, again with CRISPRa, but reading out the dozens of genes in the vicinity with targeted scRNA-seq. With this two-step strategy, we combine the scalability and unbiased nature of tiling CRISPR screens with the high-resolution targeted scRNA-seq readout.

To showcase this approach, we use it to comprehensively dissect the transcriptional regulation of an entire genomic locus, the 2 Mb regulatory landscape of PHOX2B, a master regulator of autonomous nervous system development and a key player in the development of a variety of disorders, such as the childhood cancer neuroblastoma.[22] Through an unbiased tiling screen using a robustly activating CRISPRa construct, we identified 619 CRISPRa-responsive elements (CaREs) that influence cellular survival. To identify the targets of CaREs and study the molecular effect of their activation on genes across a larger genomic context, we developed a targeted single-cell activation screen followed by sequencing (TESLA-seq). TESLA-seq combines CRISPRa screening with targeted scRNA-seq, enabling the detection of affected genes for thousands of perturbations in a single experiment. We applied TESLA-seq to the hits from the phenotypic screen to quantify their impact on the expression of transcripts within a 6 Mb space surrounding PHOX2B.

Many TESLA-seq-identified CaRE-gene interactions exhibit characteristics in agreement with current understanding of CREs, but previous evidence from neuroblastoma cell lines alone cannot fully explain the revealed functional regulatory relationships. We combined the TESLA-seq results with available epigenomic maps from 800 tissues to assign systems in which they are likely active and to define CaRE-gene pairs for which there was no previous regulatory evidence. We validate that functional transactivation can originate from elements irrespective of whether they exhibit typical regulatory traits prior to activation.

## RESULTS

### CRISPRa screening reveals candidate elements in the PHOX2B regulatory landscape

PHOX2B is a master regulator of neurogenesis and a key player in the development of neuroblastoma, whose expression level is tightly associated with the growth rates of neuroblastoma cell lines.[23–26] It has been suggested to be under the control of a large cluster of enhancers (super-enhancers),[27] and it is located in the vicinity of other genes that play roles in a variety of disorders,[22] increasing the challenge of enhancer-target prediction. Therefore, we sought to perform an exhaustive search for REs affecting growth or viability in the 2 Mb genomic space surrounding PHOX2B using CRISPRa (Figure 1A), without requiring any previous annotations or characteristics of CREs.

We tested several CRISPRa constructs for robust activation of known enhancers[28] (Figure S1A) and selected dCas9-VPR for further experiments due to it having the strongest activation and because it consists of three potent transcriptional activators that have been shown to be precise and localized.[28] We next tested whether we could successfully activate and repress PHOX2B expression and detected robust activation by targeting dCas9-VPR at its promoter in the neuroblastoma-derived cell line SHSY-5Y (Figure S1B). This action reduced cellular viability, aligning with the literature[23,24] (Figure 1B).

We conducted a dense tiling viability screen in the PHOX2B locus, designing 2–3 gRNAs per 100 bp genomic bin within the 2 Mb space (46,722 gRNAs total). These were delivered via lentivirus to SH-SY5Y-VPR cells and collected at multiple time points

(C) Volcano plot showing the log fold change of gRNA representation between the first and last time points of the experiment. Significantly enriched/depleted gRNAs (FDR < 0.05) are shown in red and blue, respectively. $p$ values shown are adjusted for multiple testing (FDR).

(D) Graphical representation of grouping strategy for analysis. Asterisks denote significance.

(E) The volcano plot shows the results of CaRE analysis, and each dot corresponds to a CaRE. The $x$ axis shows the slope calculated by our mixed linear model (MLM). Significantly enriched/depleted CaREs (FDR < 0.05) are shown in red and blue, respectively.

(F) Bar plot representing the number of CaREs, with indicated CaRE width on a log10 scale.

(G) Number of gRNAs that contribute to each CaRE, segregated by CaRE width on a log10 scale.

(H) Number of significant intragenic, intergenic, and promoter-proximal CaREs.

(I) CaREs signal around the PHOX2B locus (±1 Mb). From bottom to top: annotation for PHOX2B and its location within the genome, ChIP-seq signal for HAND2, PHOX2B, H3K27ac, and ATAC-seq signal in SH-SY5Y cell line. At the top is the score and direction (blue for depletion and red for enrichment) of CaREs. $p$ values shown are adjusted for multiple testing (FDR).

(J) Coefficient estimates from logistic regression, predicting CaREs by ATAC-seq and histone modification ChIP-seq signal in neuroblastoma cell line SH-SY5Y.

(K) Coefficient estimates from individual logistic regressions, predicting CaREs by neuroblastoma core regulatory circuit transcription factor ChIP-seq signals. Whiskers in (J) and (K) indicate 95% confidence interval of estimate and FDR in (J) and (K) is derived from $p$ values of Z-statistic. FDR, false discovery rate; NS, not significant. See also Figure S1 and Table S3.

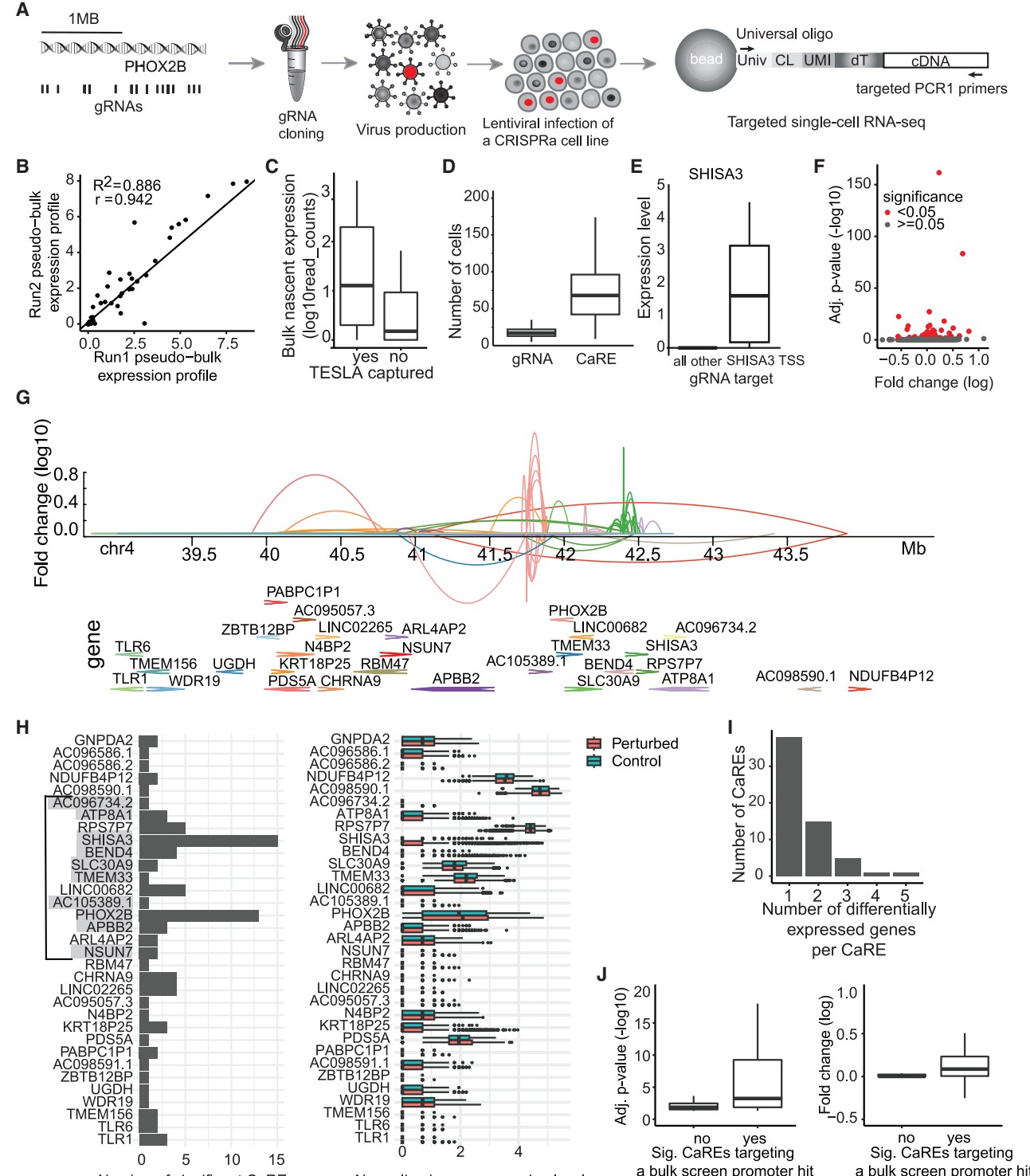

**Figure 2. TESLA-seq captures gene expression change sensitively and reproducibly**

(A) Schematic representation of the TESLA-seq.

(B) Pseudobulk expression profiles of two TESLA-seq experiments.

(C) Expression of genes detected by bulk chromatin RNA-seq that were captured by TESLA-seq (yes) versus the ones not captured by TESLA-seq (no).

(D) Number of cells assayed in TESLA-seq per gRNA and CaRE.

(E) TESLA-seq normalized gene expression level, comparing cells having gRNA targeting SHISA3 promoter versus all other cells.

in two replicates. Deep sequencing identified gRNAs that were depleted or enriched, indicating regions influencing cellular growth/viability upon activation (Figure 1A). This approach identified 758 (1.7%) depleted and 27 (0.1%) enriched gRNAs that, respectively, repress or promote cellular growth/viability upon dCas9-VPR activation (false discovery rate [FDR] < 0.05) (Figure 1C; Table S3). Positive control gRNAs targeting the PHOX2B promoter were depleted, confirming that overexpression reduces viability (Figure S1C).

To define REs from individual hits, we applied a sliding window approach to identify neighboring significant gRNAs, which were required to have the same effect on the cellular phenotype to be merged into individual functional regions. In this way, we *de-novo*-identified CaREs (Figure 1D) across the whole locus, with 536 CaREs showing gRNA depletion (genomic regions whose activation leads to a reduction of cellular viability) and 83 CaREs showing gRNA enrichment (genomic regions whose activation leads to an increase in cellular growth/proliferation; FDR < 0.05; Figure 1E; Table S3). The average width of a CaRE is 447 bp (Figure 1F), the average number of gRNAs per CaRE is 8.3 (Figure 1G), and most CaREs are promoter distal (Figure 1H). The highest-scoring CaREs are in the proximity of the PHOX2B promoter. They overlap a super-enhancer predicted to regulate PHOX2B expression in neuroblastoma[27] (Figure 1I). Besides PHOX2B, the promoters of several other genes coincide with CaREs that led to a reduction of cellular growth and survival: RP11-227F19 (non-coding RNA that shares a promoter with PHOX2B), AC105389, APBB2, ATP8A1 and AC096734 (which share a promoter), BEND4, DCAF4L1, NSUN7, AC105389, AC108210, AC024022, SHISA3, SLC30A9, TMEM33, UCHL1 and UCHL1.AS1 (which share the promoter), Y_RNA and AC105389.1. Some of them are known to affect cellular growth/proliferation: SHISA3 is a known tumor suppressor,[29–31] APBB2 plays a role in Alzheimer's disease[32,33] and the cell cycle,[34,35] and UCHL1 promotes cellular proliferation in cancer.[36,37] Therefore, CaREs are not limited to effects on PHOX2B; our tiling screen identified CaREs targeting any of these genes surrounding PHOX2B and, potentially, genes outside of the 2 Mb screening window.

Next, we sought to characterize the features of the CaREs we identified. Since CREs tend to reside in open chromatin regions flanked by post-translational modified histones and bound by transcription factors,[1,2,5] we generated assay for transposase-accessible chromatin sequencing (ATAC-seq) libraries and compiled available chromatin immunoprecipitation (ChIP)-seq data of neuroblastoma transcriptional core regulators in a variety of cell lines.[27,38–41] 28% of CaREs intersect with these typical enhancer features, and logistic regression models (see STAR Methods) predicted CaREs based on accessible chromatin and the histone modifications H3K27ac and H3K4me1 (Figure 1J) significantly better than random baseline models (Figures S1D–S1H; mean ROC-AUC: 0.60). Further, key neuronal transcription factors MYCN, HAND2, ISL1, ASCL1, GATA3, and GATA2 emerged as significant predictors when incorporated into the model (Figure 1K). We compared CaREs with evidence in neuroblastoma to those with evidence in other cell lines as described in the EpiMap database and found that the ones with evidence in neuroblastoma are more significant than the ones without evidence in neuroblastoma (Figures S1I–S1K).[42]

In summary, using CRISPRa screening in a neuroblastoma cell line in the 2 Mb genomic space on chromosome 4 surrounding the PHOX2B gene, we have identified 619 CaREs that play a role in cellular growth or survival. Some CaREs display typical enhancer features, such as accessibility and H3K27ac modification, but a substantial fraction do not, highlighting the importance of unbiased tiling screens to understand the regulation of gene expression in greater depth. We performed an equivalent CRISPRi screen with the same 46,722 gRNAs, but here, we focus on the CRISPRa screen, and additional data are provided in supplemental note 1 (Table S4).

## TESLA-seq sensitively and reproducibly detects functional regulatory relationships

To determine the precise regulatory targets of the CaREs identified in the phenotypic screen, we developed TESLA-seq. TESLA-seq allows the detection of differential expression of selected transcripts for thousands of perturbations in a single experiment by combining CRISPRa screening with a targeted scRNA-seq readout. To perform TESLA-seq, we selected 1,046 gRNAs corresponding to the top 222 CaREs from our phenotypic CRISPR viability screen and 52 gRNA controls (Figure 1; see STAR Methods). We synthesized and cloned them into a CRISPR droplet sequencing (CROP-seq) vector (Figure 2A) that enables a direct readout of the gRNA from a polyadenylated transcript in scRNA-seq experiments.[13] Next, we infected SH-SY5Y-VPR cells at a low multiplicity of infection (MOI) to ensure that each cell would only receive one gRNA. Four days post-transduction, we assayed 20,000 cells in each of two experiments on the BD Rhapsody Express Single-Cell Analysis system. In this microwell-based system, the polyadenylated transcriptome of each cell is captured using barcoded magnetic beads and subsequently amplified by two sets of primers (universal and gene specific) for each transcript in semi-nested multiplex PCRs. This is followed by sequencing

---

(F) Adjusted *p* values (−log10) relative to fold change (log2) of each CaRE-gene regulatory pair.

(G) Genome browser snapshot representing genomic distance and regulatory relationship between a CaRE and a gene determined by TESLA-seq. Each link represents an effect of a CaRE on the gene. Color reflects the target gene.

(H) Number of significant CaREs that cause a differential gene expression of an indicated number of genes. Highlighted genes are the ones whose promoters were significant hits in the bulk screen. Genes are displayed in the order that matches their order in the genome in decreasing genomic coordinates. The line indicates the screening window of the phenotypic CRISPRa screen from Figure 1.

(I) Histogram showing a number of genes differentially expressed by a perturbation of an indicated number of CaREs (left) and a histogram showing normalized gene expression level in perturbed and control cells (right).

(J) Boxplots of adjusted *p* values (−log10) (left) and fold change (log2) (right) of CaRE-gene regulation pairs. Grouped by whether the CaRE belongs to bulk screen promoter hit or not.

See also Figure S2 and Tables S5 and S6.

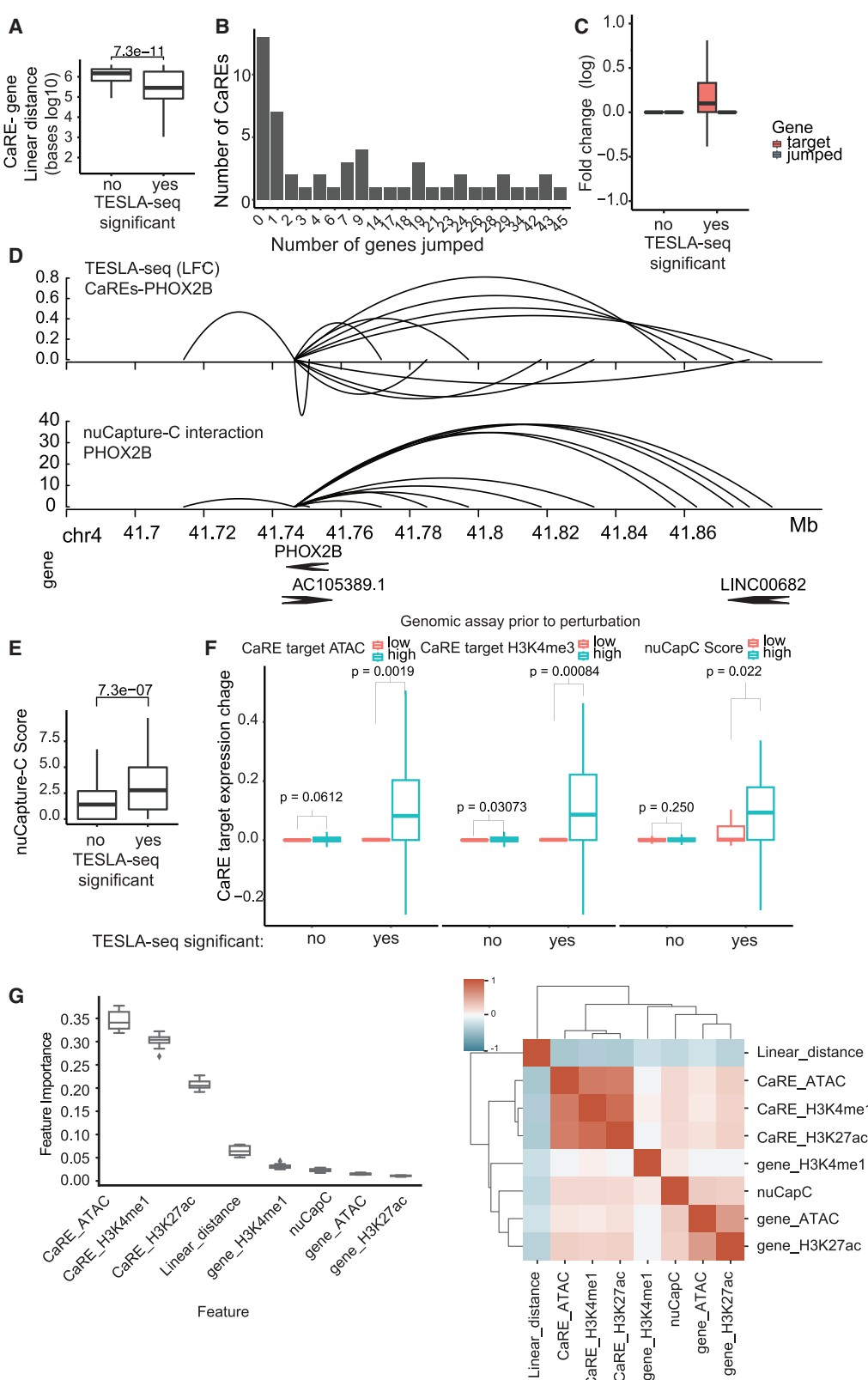

upon the addition of Illumina indices (Figure 2A).[43] Upon retrieval of barcoded complementary DNA, we enriched for the gRNA transcripts and transcripts of interest, in our case, 151 transcript isoforms annotated for all 78 annotated genes in an expanded 6 Mb space surrounding PHOX2B to detect possible effects on genes outside of the tiled screening window (see STAR Methods). In our experiments, 93% of the reads are mapped to the enriched targets.

We applied stringent quality filtering, after which 16,535 cells were retained from two BD Rhapsody runs (see STAR Methods; Figure S2A). The scRNA-seq readouts of the two TESLA-seq experiments are highly correlated (Pearson correlation $R^2$ = 0.886; Figure 2B), indicating excellent reproducibility. We capture gRNA in 99% of cells with a minimum of 2 molecules/cell, and 59 out of 78 genes within the 6 Mb space surrounding PHOX2B are captured. This capture efficiency is comparable to the capture efficiency reported for targeted Perturb-seq (TAP-seq).[17] Non-captured genes are expressed at relatively low levels in a bulk chromatin RNA-seq assay (Figure 2C). Each cell has a median of 1 unique gRNA as aimed for by the experimental design (Figure S2B). On average, each gRNA or CaRE is assayed in 17 or 68 cells, respectively (see STAR Methods; Figure 2D). These results demonstrate that TESLA-seq efficiently captures both the gRNA transcripts and the transcripts of interest with high sensitivity and reproducibility.

We utilize the targeted single-cell transcriptome replicates upon activation of each CaRE for differential gene expression analysis by comparing the expression of transcripts in the cells carrying the gRNA of interest to all other cells containing either control or distal gRNAs (at least 4 kb from the examined gRNA). As proof of principle, we focus on the expression of PHOX2B, BEND4, and SHISA3. We observe that cells in which the gRNAs fall within their promoter exhibit higher expression than the cells carrying control gRNAs (Figures 2E and S2C), demonstrating that TESLA-seq can be used to sensitively detect differential gene expression.

Next, we examined the effect of all CaREs and identified 60 CaREs that cause a significant expression change of 33 of the captured transcripts, for a total of 92 CaRE-gene pairs (adjusted p [adj. p] < 0.05) (Figures 2F–2H and S5; Table S5). On average, each CaRE causes a differential expression of 1.5 genes, with 22 of them affecting more than one gene (Figure 2I). In turn, each examined gene is significantly affected by an average of 2.8 assayed CaREs (Figures 2H and S2D). Of note, we only examined a subset of CaREs in the 2 Mb

screening window and their effects on genes located within a 6 Mb window. This leaves the possibility of additional CaREs regulating assayed genes within this window and beyond. Notably, most of the TESLA-assayed CaRE target genes whose promoters were hits in the bulk growth screen are targets in the TESLA-seq, and these interactions are especially strong and significant (Figures 2H, highlighted genes, and 2J). This is expected since the TESLA-seq guides were chosen based on their phenotypic effect in the tiling bulk screen (Figure 1), and these results can, therefore, be considered as validation for CaREs affecting growth/viability. Overall, we observe comparable results when we perform similar analyses at the gRNA rather than the CaRE level (Figures S2E–S2G; Table S6). Some of the CaREs coinciding with promoters also alter the expression of other genes, potentially indicating that these promoters can also act as enhancers.[6,44,45] In summary, our findings demonstrate that TESLA-seq can sensitively detect potential transcriptional REs and their targets.

### TESLA-seq CaRE-target interactions exhibit properties consistent with current models of CREs

To gain further insight into the identified CaREs and their targets, we explored the distance between them in both linear and 3D genomic space. Significant CaRE-gene relationships are closer in the linear genome (median distance of 284 kb; Figures 3A and S3A) than non-significant CaRE-gene relationships. In 71% (65) of significant cases, there is at least one gene between a CaRE and its target (Figure 3B), with an average of 7.7 skipped genes. These results are also consistent when the analysis is done separately on coding and non-coding genes (data not shown). Of these, the strongest hits tend to be closer to their CaREs, with up to 5 skipped genes (Figure S3B). However, the expression of these skipped genes is unaffected by the CaREs (Figure 3C), demonstrating the precise targeting of the CRISPRa construct and the ability of TESLA-seq to characterize distal regulatory relationships.

To assess the proximity of CaREs to their target genes in 3D space prior to CRISPR perturbation, we captured all 3D promoter interactions in the 6 Mb PHOX2B genomic space in the SH-SY5Y cell line using nuclear capture C (nuCaptureC[46]; Figure 3D; see PHOX2B promoter interaction example in Figure S3C). NuCaptureC interaction scores for significant CaRE-gene pairs are higher than for non-significant CaRE-gene pairs (Figure 3E). Significant CaRE-gene pairs displaying prior 3D interaction result in greater changes in expression levels

**Figure 3. Properties of CaREs and their targets identified by TESLA-seq**

(A) Linear distance between CaREs and their targets (bases in a log10 scale) stratified by their significance in the TESLA screen (p adj. < 0.05 is yes).

(B) Histogram showing the number of genes located between a CaRE and a gene in the genome (jumped genes) for each CaRE-gene regulatory pair.

(C) Comparison of the fold change (log2) of targeted genes and jumped genes between significant and non-significant TESLA-seq CaRE-gene regulation pairs.

(D) Genome browser snapshot of CaREs affecting PHOX2B expression determined by TESLA-seq (top) and PHOX2B 3D interactions determined by nuCaptureC (bottom).

(E) Comparison of nuCaptureC score between significant and non-significant TESLA-seq CaRE-gene regulation pairs.

(F) Comparison of the CaRE target gene expression (average fold change at log2 scale) between TESLA-seq significant and non-significant CaRE-gene regulation pairs that have a low or high ATAC signal at the promoter of the target gene (left), H3K4me3 signal at the promoter of the target gene (middle), and nuCapC score (right).

(G) Features that contribute the most to the predictability of a random forest model. The model predicts whether a CaRE-gene regulation pair is in the top 50 or the bottom 50. Right: hierarchical clustering of features based on correlation.

All indicated p values were calculated using the two-sided Wilcoxon rank-sum test. See also Figure S3 and Tables S5 and S6.

compared to significant CaRE-gene pairs that are not in prior 3D proximity (Figures 3D and 3F, right). Furthermore, TESLA-seq contains data for over 1,000 CaRE-gene relationships that show prior interaction according to nuCaptureC data but do not affect the interacting gene's expression when targeted by CRISPRa (Figure 3F, right). These observations suggest that prior 3D proximity allows for stronger transactivation but is neither required nor sufficient.

Our observations are further supported by published databases of regulatory associations determined either via correlations of histone modifications across cell types[42] (Figures S3D and S3F) or from high-throughput chromosome conformation capture (Hi-C) catalogs[47] (Figures S3E and S3F). Of note, only a minority of identified CaRE-gene pairs have reliable evidence in these 3D- or correlation-based regulatory associations, highlighting a potential advantage of our functional relationships over other indirect approaches.

CaRE-gene pairs for which the target gene promoters displayed high signals for ATAC and H3K4me3 showed greater changes in target gene expression upon CaRE activation compared to pairs with low levels of these chromatin features at the target promoter (Figures 3F and S3G). We trained a simple yet highly predictive random forest model for significant CaRE-gene pairs based on the genomic features mentioned above. We find that linear distance, the presence of active histone marks (H3K4me1 and H3K27ac), and accessibility are the most predictive features for regulatory relationships, with perfect separation between the top and bottom 50 CaRE-gene pairs (see STAR Methods; Figure 3G). Results are very similar when we perform logistic regression or compare individual features between significant and non-significant CaREs (Figures S3H and S3I).

In summary, many TESLA-seq-identified CaRE-target interactions exhibit characteristics consistent with current models of CREs, but as expected for the gain-of-function approach, evidence from SH-SY5Y prior to CRISPRa cannot fully explain the revealed functional regulatory relationships.

### TESLA-seq identifies regulatory network edges normally active inside and outside of the experimental system

Utilizing CRISPRa allows for the discovery of not only CREs active in the assayed cell line but also, in principle, for the discovery of genetic REs that may be active in any biological context.[48] Therefore, we examined EpiMap data, in which a compendium comprising 10,000 epigenomic maps across 800 samples is used to define chromatin states, accessible regions, promoters, enhancers, and target genes.[42]

Out of 60 CaREs participating in significant TESLA-seq interactions, 52 are defined by EpiMap as enhancers (30), promoters (7), or accessible regions (15) (Figures 4A and S4A). For all of these, we provide the target gene as identified by TESLA-seq as well as the tissue in which the genetic RE is annotated as endogenously active in EpiMap (Table S5). In 31 cases, the EpiMap-annotated target gene is in agreement with the TESLA-seq, validating both the TESLA-seq experiment and the EpiMap computational prediction.

We classified all CaREs having a significant effect on at least one gene into three groups according to regulatory evidence in

published chromatin accessibility or ChIP-seq data: 31 CaREs (involved in 51 interactions) have regulatory evidence in examined neuroblastoma (nb) cell line data, 21 CaREs (involved in 29 interactions) have evidence in EpiMap data only (other), and 8 CaREs have no previous evidence (12 interactions; no evidence found [noEF]) (Figures 4A, 4B, and S4A). We then compared these three classes of CaREs to determine if there are any striking differences. Laying out the results per gene (Figure 4A), we observed that (1) some genes have no regulators despite being surrounded by genes with altered expression in response to CRISPRa (as described in Figures 3B and 3C), (2) many interactions in the "nb" group are closer in linear or 3D space (quantified in Figures 4E, 4F, S4B, and S4D–S4F), and (3) most of the genes targeted by CaREs in the "other" or "noEF" classes are also targeted by CaREs in the "nb" class (quantified in Figure S4C). Of note, the "noEF" and "other" classes cannot be explained purely based on prior 3D proximity, as there are a number of functional interactions despite a lack of prior 3D interaction. Considering the most significant target per CaRE, we find differences between the groups for their TESLA-seq significance (adj. $p$) and distance to target gene but no difference between the groups for their effect size (log fold change) on their target genes (Figures 4C–4E and S4B), arguing against borderline false positives due purely to the chosen statistical cutoff (adj. $p < 0.05$).

Together, these results suggest that TESLA-seq can activate and identify targets for REs active both within and outside of the assayed cellular context and that specific and functional transactivation can occur from elements with or without evidence of known regulatory characteristics, with or without prior 3D proximity. Furthermore, combining epigenomic databases with TESLA-seq results can associate CaRE-gene relationships with the tissues in which they are endogenously active.

### TESLA-seq identified regulators of APBB2 and SHISA3

In addition to PHOX2B, our initial focus for identifying CREs in the CRISPRa screen, TESLA-seq has identified CREs of several other genes that play important roles in both health and disease. We examined two examples in more detail.

APBB2 plays a role in Alzheimer's disease[32,33] and the cell cycle.[34,35] TESLA-seq identified 3 CaREs for this gene (Figure 5A). CaRE_233 is the promoter of the APBB2 gene (highlighted). For CaRE_52, there is prior evidence in neuroblastoma epigenomics data ("nb"), and we validated it in SH-SY5Y cells using individual gRNAs (Figure 5A middle). For CaRE_174, we found no evidence in neuroblastoma data, but there is evidence that it acts as an RE in cardiac tissue.[42]

By promoting the degradation of β-catenin, SHISA3 contributes to the suppression of tumorigenesis, invasion, and metastasis.[29–31] The silencing of this gene has been observed in a variety of different cancer types, such as colorectal cancer,[49,50] nasopharyngeal carcinoma,[51] lung adenocarcinoma,[52] and breast cancer.[30] TESLA-seq identified 15 CaREs that regulate SHISA3 expression (Figure 5B). For 2 of them, we do not find any evidence in either neuroblastoma or other tissues (Table S5). We validated 5 of them, including one without other evidence (Figure 5Bf), two with evidence in other contexts (Figures 5Bk and 5Bn), and two annotated neuroblastoma CREs with high (Figure 5Bm) and low fold change in the

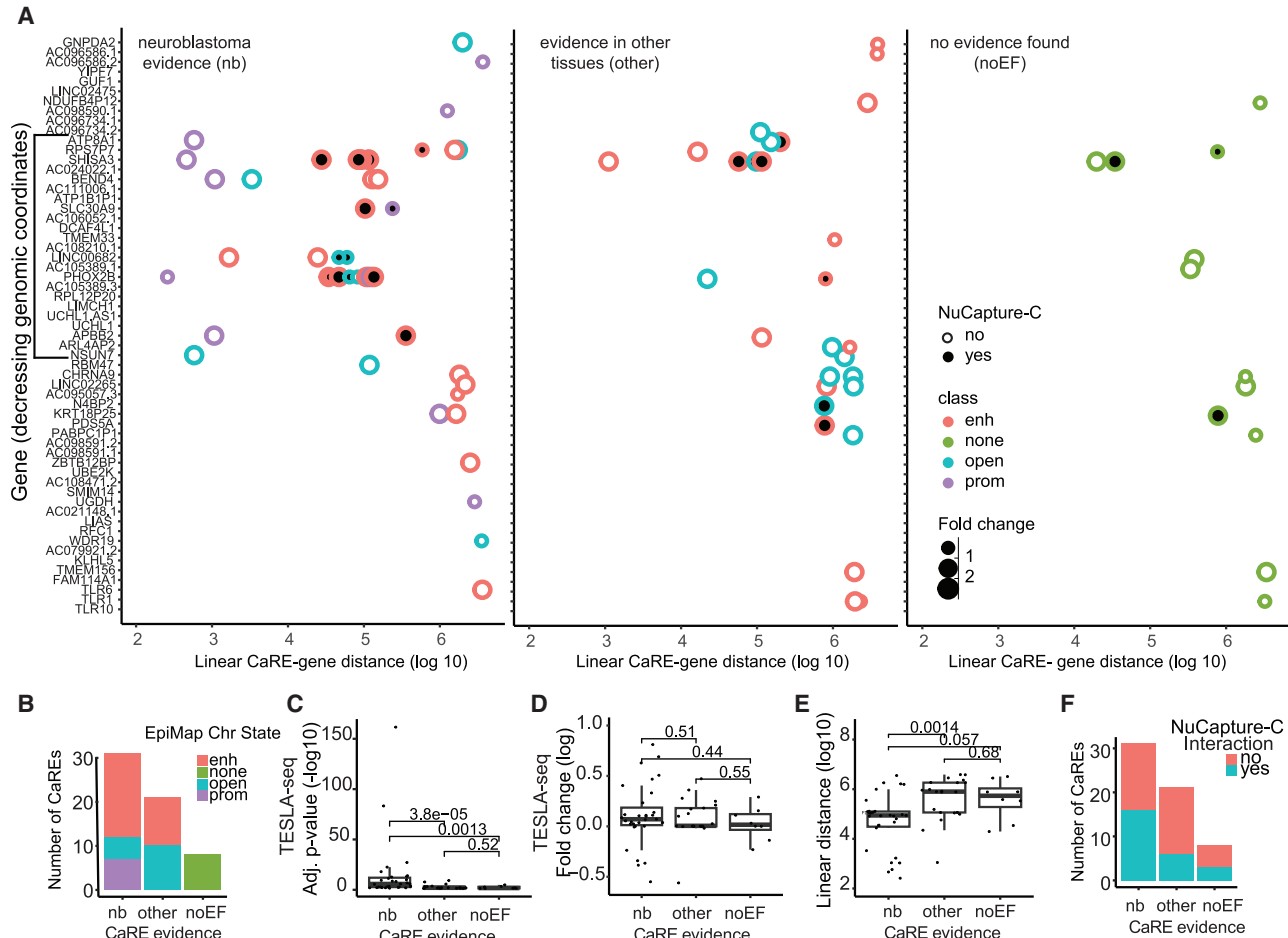

**Figure 4. Integration of TESLA-seq results with epigenomic data**

(A) For each gene target (displayed in the order that matches their order in the genome in decreasing genomic coordinates), the linear distance to their CaRE determined by TESLA-seq is shown. The line indicates the screening window in the bulk screen from which CaREs were selected for TESLA-seq. The data are further stratified by nuCaptureC signal displayed via circle fill (yes: 3D interaction is detected or no: no interaction detected); class defined by EpiMap indicated via circle color; and fold changed determined via TESLA-seq depicted by circle size. CaREs are classified into three groups according to regulatory evidence in published ATAC- or ChIP-seq data: (left) CaREs with evidence in neuroblastoma (nb $n$ = 31 involved in 51 interactions), (middle) evidence in tissue other than neuroblastoma (other $n$ = 21 involved in 29 interactions), and (right) CaREs for which no evidence was found (noEF $n$ = 12).

(B) Number of CaREs in each category, nb, other, or noEF, grouped by the EpiMap classification indicated with colors.

(C–E) Comparison between different CaRE evidence categories (nb, other, or noEF) in CaRE-gene pair adj. $p$ ($-$log10 scale) (C), fold change (log2 scale) (D), and linear distance (E).

(F) Number of CaREs in each category (nb, other, or noEF) grouped by NuCaptureC-detected interaction (yes: 3D interaction is detected or no: no interaction detected).

All indicated $p$ values were calculated using the two-sided Wilcoxon rank-sum test. See also Figures S4 and S5 and Tables S5 and S6.

TESLA-seq (Figure 5Bb). Of note, the fold change effects detected via TESLA-seq resemble the fold changes detected via RT-qPCR, highlighting the sensitivity of the TESLA-seq.

Two of the CaREs identified to regulate SHISA3 by TESLA-seq (SHISA3 promoter CaRE_790 [Figure 5Bh] and CaRE_816 [Figure 5Bk]) intersect with the somatic risk variants found in cancer.[53] In EpiMap, CaRE_816 is annotated to be active in renal tissue and cancer. This is another example of linking disease-relevant CREs to their target genes in tissues other than the one used for the screen.

In addition to regulating SHISA3, the promoter CaRE of SHISA3 (CaRE_790) is also identified to regulate the UGDH

and TLR6 genes. This could be an example of a promoter acting as an enhancer, but we cannot rule out a secondary effect of modulating SHISA3 expression.

## DISCUSSION

A wide variety of resources have cataloged CREs and their potential involvement in disease based on their biochemical features.[41,42,53] CRE validation, functional characterization, and identification of their target genes represent major challenges in the field. Here, we developed a strategy that allows the identification of CREs and their targets at a

**A**

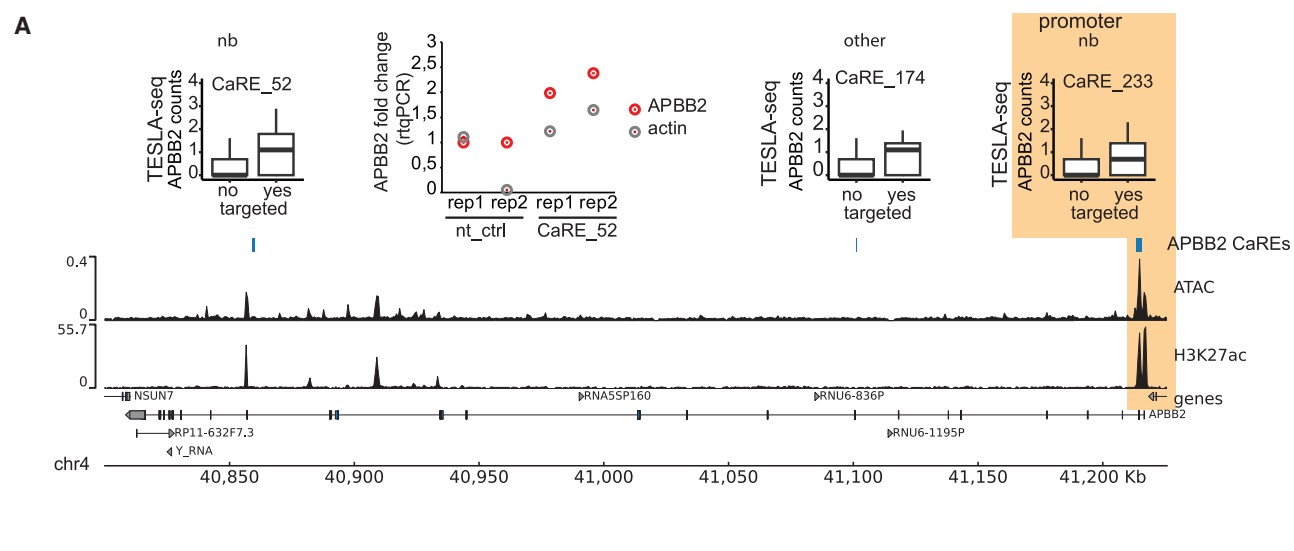

**B**

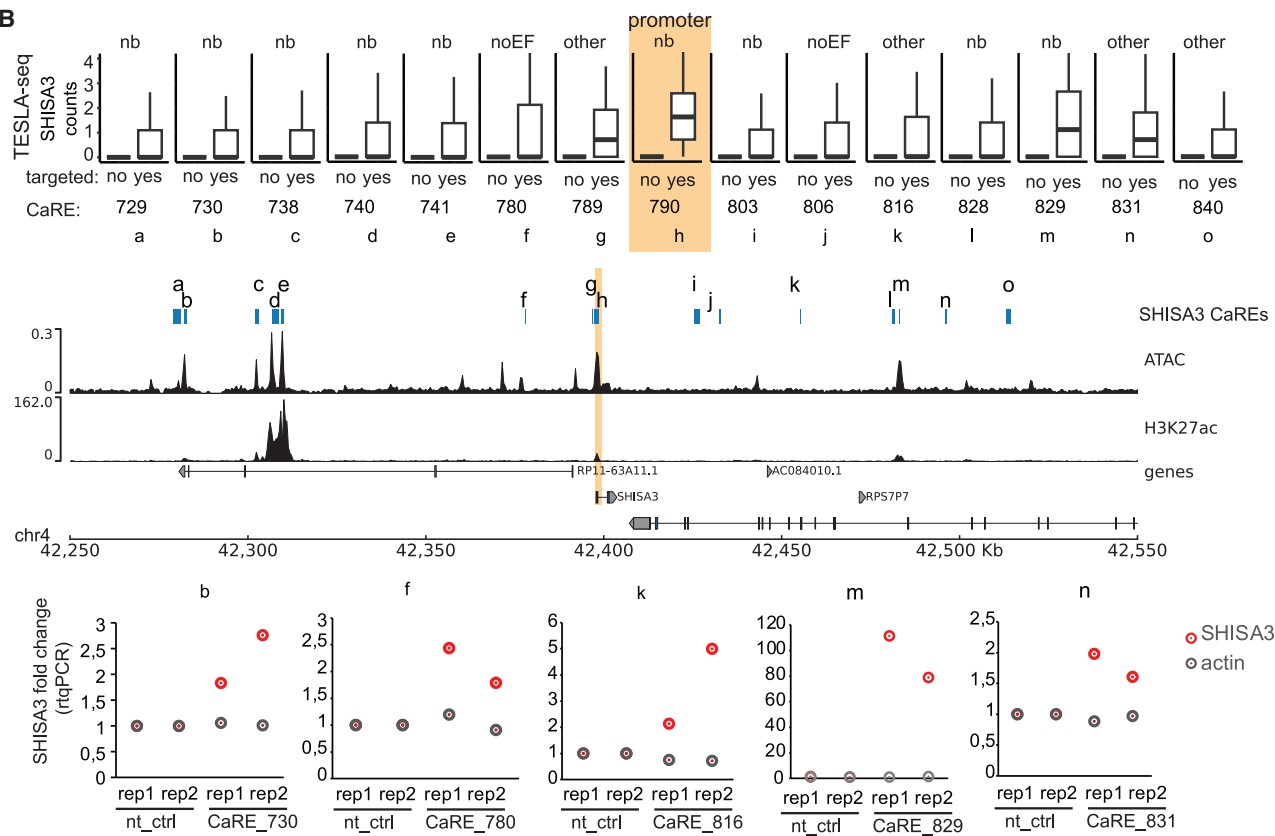

**Figure 5. Selected examples of CaRE-gene pairs**

(A) Top left and right: TESLA-seq normalized gene expression level of APBP2 gene, comparing cells having gRNA targeting individual indicated CaREs versus all other cells. Top middle: APBP2 expression change upon activation with dCas9-VPR with control or gRNA targeting CaRE_52 in SH-SY5Y cell line. Relative expression of target genes is determined by RT-qPCR and normalized to GAPDH. Shown are two replicates and actin as a control gene. Bottom: browser shot of the APBP2 genomic region. From top to bottom, the following tracks are displayed: TESLA-seq-identified CaREs affecting APBP2 expression, ATAC-seq and H3K27ac ChIP-seq from SH-SY5Y cell line, and gene track.

(B) Top: TESLA-seq normalized gene expression level of SHISA3 gene, comparing cells having gRNA targeting individual indicated CaREs versus all other cells. Middle: browser shot of the SHISA3 genomic region. From top to bottom, the following tracks are displayed: TESLA-seq-identified CaREs affecting SHISA3 expression, ATAC-seq and H3K27ac ChIP-seq from SH-SY5Y cell line, and gene track. Bottom: SHISA3 expression change upon activation with dCas9-VPR with control or gRNA targeting indicated CaRE in SH-SY5Y cell line. The following gRNAs were used: X886 for CaRE_730, X327 for CaRE_780, X341 for CaRE_816, X351 for CaRE_829, and X355 for CaRE_831. Relative expression of target genes was determined by RT-qPCR and normalized to GAPDH. Shown are two replicates and actin as a control gene.

large scale and at high sensitivity by combining an unbiased tiled phenotypic screen with TESLA-seq—a CRISPRa screen with a targeted scRNA-seq readout of guides and their effects.

In our application on dissecting the regulatory landscape of the PHOX2B locus, we set up a tiling CRISPRa screen in a 2 Mb window and identified 619 CaREs that affect cellular growth/proliferation. For a subset of 222 CaREs, we examined the effect on the expression of genes within an extended 6 Mb window via TESLA-seq, resulting in 92 functional CaRE-target gene pairs. Since we selected CaREs based on a phenotypic effect in the tiled screen window, there are likely many functional CaREs in this window that we did not assay, affecting genes within or outside of this window. Due to the novelty of the approach, we employed very stringent and conservative filtering criteria at every step of the TESLA-seq analysis. Additional data from further TESLA-seq will allow us to improve on computational tools for the analysis of this type of data.

On average, there are 7.7 genes skipped between a CaRE and its target, meaning that the commonly used strategy of associating an enhancer with its closest active gene is likely suboptimal,[5,54] although this will strongly depend on the specific locus. These observations underline the difficult task of computationally predicting enhancers and their targets and the need for scaling up the functional characterization of CREs. Since the expression of skipped genes is not affected (Figure 3C), these results also demonstrate the sensitivity of TESLA-seq to characterize distal regulatory relationships. However, due to the time period elapsing between transduction and single-cell readout, we cannot rule out additional secondary/indirect regulatory effects. While further studies are needed to confidently distinguish between the two, the observed evidence for prior 3D interactions between significant CaRE-gene pairs (Figures 3F, right, and 4A), as well as the overlap with known REs (Figures 4A and S4A) and targets, suggests that we are primarily identifying direct effects (see also supplemental note 2).

Given the current understanding about CREs,[5,54] we find that the majority of significant enhancer-gene pairs interact in 3D genomic space before CRISPRa. Prior 3D proximity allows for stronger transactivation; however, it is neither sufficient nor required (Figures 3F, right, and 4A). This observation further exemplifies that, although 3D proximity can be an indicator of a functional relationship between a CRE and a gene, functional characterization is necessary to reliably assign a regulatory effect of a CRE on its target.

A major advantage of an activation screen is that it includes CREs that are not necessarily active in the assayed cell line. By combining the TESLA-seq results with available epigenomic maps from 800 tissues, we were able to assign the likely relevant tissue for 29 CaRE-gene relationships. Using a CRISPR activator to assay possible CREs regardless of their endogenous activity is especially important for studying CREs that are only active in rare cell types and cell types that we cannot easily culture and study.

Finally, the epigenetic landscape at the time of the experiment will influence the efficiency of CRISPR perturbation. In general, CRISPRi/a tends to be more efficient in open chromatin regions,[55] and although gene activation by dCas9-VPR is successful in most genomic contexts, including bivalent chromatin, constitutive heterochromatin is less responsive.[48] Our findings are in line with this: the most responsive CaREs are the ones that are in a permissive chromatin state in the assayed cell line. These results may reflect a combination of dCas9 requiring physical accessibility to its DNA target and/or dCas9-VPR requiring other CRE features, such as well-positioned nucleosomes and divergently oriented core promoter sequences.[56–58] To bypass these limitations, future studies will benefit from the use of other activators and repressors as well as screening in a handful of different cell types.[8,54,59]

TESLA-seq enables detailed, sensitive investigation of gene expression regulation. It is a stand-alone approach and can be used to directly read out perturbation effects without any phenotypic preselection, as described in this manuscript. Its high-throughput approach offers the potential to uncover transcriptional regulatory logic in its natural environment at a competitive scale, and it provides a significant step forward for uncovering the effects of non-coding sequence variation on disease.

## Limitations of the study

This study is primarily limited by the following factors. (1) Cell line selection: although we used a CRISPR activator that allows us to study elements that are not necessarily active in the assayed cell line, the effectiveness of the CRISPR construct still depends on the cellular context, e.g., elements located in a tightly packed heterochromatin may not be responsive to CRISPR-mediated activation. (2) The choice of a CRISPR construct: while dCas9-VPR is a potent activator, it may not effectively activate all true REs. Using different constructs with different modes of action, including the wild-type Cas9 (wtCas9) and various activators and repressors, would provide a more comprehensive understanding. (3) Readout of a subset of transcripts: in order to increase the sensitivity, we have only examined the genes within the 6 Mb genomic space surrounding the PHOX2B gene. Although this increases the sensitivity of TESLA-seq, we aim to read out the whole transcriptome at that same level of sensitivity as scRNA-seq technologies develop. Future studies may broaden our insights via multiple cell lines, organoids, and animal models with a variety of different CRISPR constructs to dissect the regulatory architecture of the whole genome and its effect on the whole transcriptome and proteome.

## RESOURCE AVAILABILITY

### Lead contact

Further information and requests for resources and reagents should be directed to and will be fulfilled by the lead contact, Uwe Ohler (uwe.ohler@mdc-berlin.de).

### Materials availability

Correspondence and requests for materials should be addressed to the technical contact, Dubravka Vučićević (vucicevic.dubravka@gmail.com). The unique identifiers of all biological materials are listed in the STAR Methods. The newly generated plasmids have been deposited to Addgene: pPB-CAG-dCas9-VPR-pgk-hph (Addgene #233066) and PPB-CAG-KRAB-dCas9-P2A-mCherry-pgk-hph (Addgene #233067).

### Data and code availability

- Data generated for this study are accessible at GEO under accession numbers GEO: GSE274254 for ATAC-seq, GSE274255 for bulk CRISPR screen, GSE274256 for TESLA-seq, GSE274257 for nuCaptureC, and GSE274258 for chromatin RNA-seq.
- All codes for data analysis and visualization have been archived at Zenodo, DOI: https://doi.org/10.5281/zenodo.15719546, are mirrored on GitHub at https://github.com/ohlerlab/Vucicevic_et_al, and are publicly available.
- Any additional information necessary to re-analyze the data reported in this paper is available from the technical contact upon request.

### ACKNOWLEDGMENTS

We thank Marieke Oudelaar for advice on nuCaptureC and for sharing unpublished work with us. We thank the MDC and the Helmholtz association for funding, including its MDC/BIMSB-NYU exchange program. This work was further supported by International Research Training Group 2403 and Research Unit 2841 of the Deutsche Forschungsgemeinschaft.

### AUTHOR CONTRIBUTIONS

D.V. performed all experiments with technical assistance from A.H. and I.B., performed some bioinformatic analysis, and made the figures. L.S.L.Z., M.B., and S.A.L. performed the bioinformatic analysis of the bulk CRISPR screens. C.-W.H. and S.A.L. performed the bioinformatic analysis of the TESLA-seq data. N.K. and M.L. cloned the dcas9-VPR and dCas9-KRAB piggyback vector. D.V., S.A.L., and U.O. wrote the manuscript. C.-W.H., L.S.L.Z., and M.B. gave comments and suggestions on the manuscript and performed minor editing. D.V., S.A.L., and U.O. designed the study.

### DECLARATION OF INTERESTS

The authors declare no competing interests.

### STAR★METHODS

Detailed methods are provided in the online version of this paper and include the following:

- KEY RESOURCES TABLE
- EXPERIMENTAL MODEL AND STUDY PARTICIPANT DETAILS
- METHOD DETAILS
  - Cell culture
  - Comparison of CRISPR activators
  - Cloning and transduction of individual gRNAs
  - Cloning of dCas9 constructs
  - Generation of stable cell lines
  - Viability assays
  - qPCR measurement of CRISPRa/i effects
  - Design and selection of gRNAs for the bulk screen
  - CRISPRa/i viability screen
  - Bulk CRISPRa/i data analysis
  - ATAC-seq
  - ATAC-seq processing
  - Chromatin RNA-seq
  - RNA-seq processing and analysis
  - Analysis of Ca/iRE features in neuroblastoma data
  - TESLA-seq
  - Computational analysis of TESLA-seq data
  - Nuclear capture C
  - Nuclear capture C data processing
  - Hi-C data processing
  - Random forest model for predicting significant CaRE-gene pairs from genomic features
  - TESLA-seq multimodal integration analysis
  - Supplemental note 1 - CRISPRi screen
  - Supplemental note 2 - Characteristics of CRISPR-mediated activation
- QUANTIFICATION AND STATISTICAL ANALYSIS

### SUPPLEMENTAL INFORMATION

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

**Cell Genomics**
Article

# STAR★METHODS

## KEY RESOURCES TABLE

| REAGENT or RESOURCE | SOURCE | IDENTIFIER |
|---|---|---|
| Chemicals, peptides, and recombinant proteins | | |
| Esp3I | Thermo Fisher Scientific | #FD0454 |
| PiggyBac transposase | BioCat | #PB210PA-1-SBI |
| Calcein AM | Thermo Fisher Scientific | #C1430 |
| Draq7 | Thermo Fisher Scientific | #564904 |
| Critical commercial assays | | |
| RealTime-Glo MT Cell Viability Assay | Promega | #G9711 |
| Lenti-X Concentrator | Takara Bio | #631232 |
| NextSeq 500/550 HighOutput v2 Kit | Illumina | #FC-404-2002 |
| NEXTflex Rapid Directional qRNA-Seq Kit | Bioo Scientific | NOVA-5130-01D |
| BD Rhapsody cartriges | BD Bioscience | Cat. No. 633733 |
| BD RhapsodyTM Cartridge Reagent Kit | BD Bioscience | Cat. No. 633731 |
| BD RhapsodyTM cDNA Kit | BD Bioscience | Cat. No. 633773 |
| Deposited data | | |
| TESLA-seq | This paper | GEO: GSE274256 |
| ATAC-seq | This paper | GEO: GSE274254 |
| bulk CRISPR screen | This paper | GEO: GSE274255 |
| nuCapture-C | This paper | GEO: GSE274257 |
| chromatin RNA-seq | This paper | GEO: GSE274258 |
| Code repository | This paper | Zenodo DOI: https://doi.org/10.5281/zenodo.15719546 |
| Experimental models: Cell lines | | |
| SHSY-5Y | DSMZ | #ACC 209 |
| Oligonucleotides | | |
| Table S1 | This paper | Primers and individual gRNAs oligoes |
| Table S2 | This paper | Oligoes used for nuCaptureC |
| Table S3 | This paper | gRNAs used for the phenotypic screen |
| Table S6 | This paper | gRNAs used for TESLA-seq |
| Recombinant DNA | | |
| CROPseq-Puro | Datlinger et al.[13] | addgene #86708 |
| PPB-CAG-KRAB-dCas9-P2A-mCherry-pgk-hph | This paper | #233067 |
| pPB-CAG-dCas9-VPR-pgk-hph | This paper | #233066 |
| Software and algorithms | | |
| Seurat (v4) | Satija Lab | https://satijalab.org/seurat/ |

## EXPERIMENTAL MODEL AND STUDY PARTICIPANT DETAILS

SHSY-5Y cells (DSMZ #ACC 209), were purchased authenticated and are from a female donor.

## METHOD DETAILS

### Cell culture

SHSY-5Y cells (DSMZ #ACC 209) were cultured in DMEM/F12 (Thermo Fisher Scientific #31330038) supplemented with 20% FBS (Thermo Fisher Scientific #16000044) and 1% penicillin/streptomycin (Thermo Fisher Scientific #15070063) at 37°C with 5% $CO_2$. When 90% confluent, cells were washed with PBS (Thermo Fisher Scientific #10010015), trypsinized using 0.05% Trypsin-EDTA

(Thermo Fisher Scientific #25300096) for 2 min, spun down in complete growth medium at 400g for 3 min and split 1:6 into clean TC dishes.

### Comparison of CRISPR activators

We targeted MyoD enhancers: distal regulatory region (DRR) and core enhancer (CE) as well as the MyoD promoter through co-transfection of gRNAs targeting each region and either dCas9-p300,[60] dCas9-SunTag10,[61] dCas9-SunTag24[61] or dCas9-VPR.[62] $2 \times 10^5$ HEK293 cells were co-transfected with 20ng of either equimolar pool of gRNAs or 20ng of undivudual gRNAs and 50ng of either one of the dCas9 fusion constructs with lipofectamine2000.[62] Cells were collected 48h or 96h later in the case of dCas9-p300. Relative expression was determined by qRT-PCR and normalized to gapdh, actin is shown as a control gene.

### Cloning and transduction of individual gRNAs

Ordered oligonucleotides contained sticky-end overhangs matching the Esp3I overhangs of the CROPseq-Puro (addgene #86708) digested with Esp3I (Thermo Fisher Scientific #FD0454). Sense and antisense oligonucleotides were annealed and ligated into the digested backbone using T4 DNA ligase (NEB). They were transformed in Stbl3 bacteria and the clones were validated with Sanger sequencing. Lentiviral production was done as described in Datlinger et al.[13] The lentiviral prep was concentrated using Lenti-X Concentrator (Takara Bio #631232). Cells were transduced at a low multiplicity of infection (<0.3 MOI), selected 48h post-transduction using 2ug/ml of puromycin (Sigma Aldrich #P9620) and treated as indicated in different assays.

### Cloning of dCas9 constructs

dCas9-VPR and KRAB-dCas9-P2A-mCherry constructs were designed to be flanked by NheI and NotI restriction sites and synthesized into the pEX-A2 backbone (Eurofins). To generate dCas9 piggybac transposons, pEX-A2 plasmids containing the dCas9 constructs were digested with NheI and NotI and cloned into the backbone of pPB-CAG-3xFLAG-empty-pgk-hph (addgene #48754) between XbaI and NotI sites, rendering the XbaI site destroyed. These constructs are deposited to addgene: pPB-CAG-dCas9-VPR-pgk-hph (addgene #233066) and PPB-CAG-KRAB-dCas9-P2A-mCherry-pgk-hph (addgene #233067).

### Generation of stable cell lines

Stable SH-SY5Y-dCas9VPR and SH-SY5Y-dCas9KRAB cell lines were generated using the PiggyBacTM Transposon Vector System. According to the manufacturer's instructions, $0.5 \times 10^6$ SHSY-5Y cells (passage number p10) per well of a 6-well plate were transfected using Lipofectamine 2000 (Thermo Fisher Scientific #11668019) in a 1:5 ratio of PiggyBac transposase (BioCat #PB210PA-1-SBI) to PiggyBac transposon carrying either dCas9-VPR or KRAB-dCas9-P2A-mCherry constructs, respectively. The following day, cells were selected for successful integration using 300 μg/μl of hygromycin (Invivogen #ant-hm-1) until the control cells were dead (4 days). They were re-selected before each experiment to ensure the expression of the constructs.

### Viability assays

The viability of SH-SY5Y-dCas9VPR and SH-SY5Y-dCas9KRAB cells treated with either control, six gRNAs targeting the promoter of PHOX2B, or any other indicated treatment was measured using the RealTime-Glo MT Cell Viability Assay (Promega #G9711) according to the manufacturer's instructions. Luminescence was measured 24, 48, and 72 h after the cells were selected for gRNA expression.

### qPCR measurement of CRISPRa/i effects

Total RNA was extracted from cells following the manufacturer's instructions with Trizol (Thermo Fisher Scientific #15596018). Reverse transcription was carried out using the High Capacity RNA-to-cDNA Kit (Thermo Fisher Scientific #4387406), and RTqPCR was performed using Fast SYBR Green Master Mix (Thermo Fisher Scientific #4385616) according to the manufacturer's instructions on a Real-Time PCR System (Roche lightcycler 480 II). The relative expression was calculated by normalizing to GAPDH expression levels as a control housekeeping gene. Primers used for RT-qPCR can be found in Table S1.

### Design and selection of gRNAs for the bulk screen

100bp bins of the 2MB region surrounding PHOX2B were used as input for guidescan software[63] with parameters: guidescan_guidequery (–target within -o. –output_format csv –select score -n 3) to select the top 3 guides for each bin from a pre-computed Cas9 guide database for hg38. We included 100 control gRNAs from the GeCO V2 library.[64] Suitable overhangs for the Gibson assembly were added.[65]

### CRISPRa/i viability screen

Oligo pools containing 46,722 oligos (Table S3) ordered from Twist Bioscience were cloned into the CROPseq-Guide-Puro plasmid (Addgene #86708) digested with Esp3I (Thermo Fisher Scientific #FD0454) according to.[65] The library was amplified using primers and according to the protocol from Datlinger et al.[13] and sequenced on a NextSeq 500/550 machine according to the Illumina user manual. Lentiviral production was done as described in Datlinger et al.[13] The lentiviral prep was concentrated using Lenti-X Concentrator (Takara Bio #631232) and the viral titer was determined using a Crystal violet viability assay.[66] The screen was performed as

described in Datlinger et al.[13] For each gRNA at least 1000 cells per gRNA were seeded and they were transduced at a low multiplicity of infection (<0.3 MOI) in either SH-SY5Y-dCas9VPR and SH-SY5Y-dCas9KRAB cell line. The cells were selected 48h post-transduction for the ones that received the gRNA using 2ug/ml of puromycin (Sigma Aldrich #P9620) for three days. They were kept under constant antibiotic selection with 1ug/ml of puromycin and 100ug/ml of hygromycin and collected at several time points (day 5, day 20, day 29 and day 33). DNA was extracted using the NucleoBond Xtra Maxi Plus (Macherey-Nage #7404426.50). The library preparation was done as described in Datlinger et al.[13] and sequenced on a NextSeq 500/550 according to the Illumina protocol.

### Bulk CRISPRa/i data analysis

Sequencing reads were assigned to sgRNA using a modified version of a script in Sanjana et al.[67] Mean guide-counts were obtained at each time-point and all time-points were combined in a single table. Normalized counts for a given guide were computed by dividing the number of reads mapped to that guide by the total number of reads mapped to all guides, then multiplying the result by one million. Log-fold changes were calculated for all sgRNAs to quantify their effects.

To score the effect of individual guides, a null distribution was obtained by using non-targeting sgRNAs to represent expected variability in the absence of a true effect. The statistical significance of each sgRNA's log-fold change was then assessed using pnorm() from the stats package in R, which reports the probability of observing a log-fold change as extreme or more extreme than the observed value based on the null distribution. To account for multiple hypothesis testing, *p*-values were adjusted using the Benjamini-Hochberg procedure, and sgRNAs were considered significant if the adjusted *p*-value was less than 0.05.

To evaluate hits at the bin level, we employed a sliding window strategy to jointly assess the effect of all sgRNAs targeting the bin itself as well as neighboring upstream and downstream bins. The model used to analyze log-transformed counts for each bin was defined using a traditional linear mixed model (LMM):

$$Y = X\beta + Zb + \varepsilon$$

where:

$Y$ represents the log-transformed counts of each bin,

$X$ is the fixed-effects design matrix, which includes known or controlled variables such as Time,

$\beta$ represents the fixed-effect coefficients,

$Z$ is the random-effects design matrix (including sgRNA effects),

$b$ is the vector of random effects, assumed to follow assumed to follow a normal distribution N(0, G) where $G$ is the variance-covariance matrix of the random effects

$\varepsilon$ is the residual error term, assumed to follow N(0, R) where $R$ is the variance-covariance matrix of residuals.

The model was fitted using the 'lme4'(v. 1.1.28) *R* package with restricted maximum liklihood (REML) set to FALSE.

To identify bins significantly affected by the screen, we can use this model to test for the significance of the Time variable, i.e., whether the perturbation showed a consistent effect across guides and timepoints. To this end we compared the full model to a reduced model that excluded the Time variable using a maximum likelihood ratio via ANOVA[68] as follows: The reduced model was defined as:

$$Y = Zb + \varepsilon$$

with the same normality assumptions as above.

To correct for multiple hypothesis testing, *p*-values were adjusted using the Benjamini-Hochberg method.[69]

Adjacent significant bins within 500 bp of each other were merged to CaREs using bedtools (v 2.30.0).[70] If two significant bins were separated by more than 500 bp they were regarded as as independent CaREs. A final CaRE score uses a mixed linear model (MLM) incorporating information from all gRNAs targeting the genomic region of the CaRE. We report the negative base-10 logarithm of the adjusted *p*-value.

### ATAC-seq

ATAC-seq experiments were performed in the SH-SY5Y, CLB-Ga, IMR-5, Kelly, NGP and SK-N-SH neuroblastoma cell lines using 100,000 cells according to the protocol[71] with the following modifications: transposition time was increased from 30 min to 1 h and the cell pellets were taken directly to the transposition reaction omitting the lysis step as described in Karabacak Calviello et al.[72] For all samples, 12 PCR cycles were performed, and the libraries were sequenced (2x75nt) on a NextSeq 500/550 using a HighOutput v2 Kit for 150 cycles (Illumina #FC-404-2002, discontinued).

### ATAC-seq processing

ATAC-seq reads were trimmed for adapter content using flexbar (-f i1.8 -u 10 -ae RIGHT -at 1.0;[73]), mapped to hg19 using bowtie2 (-X 1500 –no-discordant; reads eventually lifted over to hg38; see below,[74]; filtered for unique mapping reads (grep -v "XS:i:"), and collapsed for PCR duplicates using Picard Tools MarkDuplicates (http://broadinstitute.github.io/picard). Finally, the 5′ends of reads were selected and extended to account for the estimated footprint size of Tn5 on the DNA using bedtools slop (-l 15 -r 22 -s;[70,75]; 11 (12): R119). Peaks were called using JAMM (-e auto -b 100;[76]) on fragment extended reads.

## Chromatin RNA-seq

Cellular fractionation was performed in SH-SY5Y cell line according to Conrad et al.[77] Chromatin RNA were extracted using Trizol and Direct-zol RNA MiniPrep Kit (Zymo Research #R2052) according to the manufacturer's instructions. The library was prepared using NEXTflex Rapid Directional qRNA-Seq Kit (BiooScientific #NOVA-5130-01D) according to the manufacturer's instructions and paired-end sequencing (2x75nt) was performed on a NextSeq 500/550 using a HighOutput v2 Kit for 150 cycles.

## RNA-seq processing and analysis

Unique molecular identifiers (UMIs) were extracted from.fastq files using UMI-tools,[78] and reads were trimmed using fastx_trimmer from the FASTX-toolkit (http://hannonlab.cshl.edu/fastx_toolkit/). Reads were then filtered for ERCC spike-in reads and rRNA by mapping to a custom index with Bowtie 1.[79] Trimmed and filtered reads were then mapped using STAR.[80] Mapped.bam files were subjected to PCR deduplication using UMI-tools,[78] followed by conversion to.fastq and remapping with STAR to generate final mapped files and normalized coverage tracks.

## Analysis of Ca/iRE features in neuroblastoma data

Raw sequencing data from ChIP-seq analyses of histone modifications and transcription factors in neuroblastoma cell lines[27,38–41] was downloaded from the Sequence Read Archive (https://www.ncbi.nlm.nih.gov/sra) under accessions SRR3363255, SRR3363256, SRR3363257, SRR3363258, SRR3363259, SRR5249434, SRR5249436, SRR5249437, SRR5249438, SRR5249439, SRR5249440, SRR5249442, SRR5249443, SRR5249446, SRR5249447, SRR5249451, SRR5675976, SRR5675978, SRR5676027, SRR5676028, SRR5676029, SRR6451360, SRR6451361, SRR6451362, SRR7101491, SRR7101492, SRR7865946, SRR7865947 and SRR8169718 and from the ENCODE database (https://www.encodeproject.org/) under accessions ENCFF000ZPW, ENCFF000ZPZ, ENCFF000ZQF, ENCFF000ZQI, ENCFF000ZQL, ENCFF000ZQX, ENCFF072EIX, ENCFF443LZC, ENCFF458ASE, ENCFF557DAH and ENCFF615LEB. Reads were mapped to the hg38 reference genome using bowtie2 (version 2.3.4.3). Fragment lengths were estimated for each sample (and input/background) alignment by macs2 (version 2.1.1.20160309) using parameter -g 1.6e+9. From the list of estimated fragment lengths per sample, the length closest to 200 bp was selected and read mapping coordinates of that sample were extended to the respective fragment length. Read coverage of processed ChIP-seq and ATAC read coordinates were summarized in bins and regions of the CRISPRa phenotypic screen analysis respectively, using the function summarizeOverlaps from the R/Bioconductor package GenomicAlignments (version 1.22.1) with parameters mode = "Union" and inter. feature = FALSE. To capture signals of histones adjacent to CaREs the respective summary features of histone ChIP-seq samples were resized to a minimum of 1000 bp centering on the CaRE. ATAC-seq read coordinates were translated to hg38 using the function liftOver from the R/Bioconductor package rtracklayer (version 1.46.0). Read coverage per region was normalized to reads per kilobase (RPK). Peaks were called using JAMM (-e auto -b 100;[76]) on fragment extended reads, and peaks for transcription factors were further filtered to have a peak score >515.

CaREs were defined as regions with FDR <0.05 from the respective CRISPR screen analysis. ChIP-seq and ATAC-seq read coverages were normalized to z-scores in 100 bp genomic bins for downstream analysis. A unified logistic regression model (generalized linear model with binomial link function as provided by the glm function in R) was applied to predict CaREs by ChIP-seq z-scores of H3K27ac, H3K4me1, H3K4me3 and H3K27me3 and ATAC-seq z-scores in neuroblastoma cell line SH-SY5Y. We validated our modeling approach by comparing the receiver operating characteristic area under the curve (ROC-AUC) of CaRE models to that of models trained on randomly shuffled assignments between CaREs and the epigenetic features. To that end we trained $n = 50$ models for each category (CaRE and random) from 10 repetitions of a 5-fold cross validation, where in each repetition the observations were randomly assigned to the folds. We examined significance and size of model covariates in a model trained on all available data to report the relative importance of epigenetic features for CaRE prediction. Additionally, we predicted CaREs by individual logistic regression models, one for each transcription factor ChIP-seq experiment conducted in neuroblastoma cell lines, where each model was controlled for signal z-scores of the more general epigenetic features H3K27ac, H3K4me1 and H3K4me3 from ChIP-seq and ATAC-seq in cell line SH-SY5Y. We again investigated significance and size of covariates in these models to report the relative importance of transcription factor features for CaRE prediction.

## TESLA-seq

We selected 222 top-scoring significant CaREs from the bulk phenotypic screen. Theese CaREs and their gRNAs were also the top scored ones using the MAGeCK tool.[81] For each of them, we selected gRNAs with the highest fold change. These gRNAs, together with 52 control gRNAs (total of 1098 in Table S5), were ordered as oligos from Twist Bioscience, cloned, and sequenced as described for the phenotypic screen. The lentiviral production was done as in the viability screen. SHSY-5Y-VPR line was infected at <0.3 MOI and the selection with 2ug/ul of puromycin started 24h post transduction. As soon as the antibiotic control non-infected cells were dead,on day 4 after transduction, the cells were collected.

We then proceeded according to the BD Rhapsody protocol: Single Cell Capture and cDNA Synthesis with the BD RhapsodyTM Single-Cell Analysis System (Doc ID: 210966). Cells were counted on the BD Rhapsody scanner (BD Bioscience) and the viability of the cells was >85% in both experiments as determined by the BD Rhapsody scanner after staining with Calcein AM (Thermo Fisher Scientific #C1430) and Draq7 (Thermo Fisher Scientific #564904) according to the manufacturer's protocol. Next, cells were loaded into BD Rhapsody cartridges (BD Bioscience Cat. No. 633733) and incubated with the beads (BD RhapsodyTM Cartridge Reagent Kit

Cat. No. 633731). Followed by stringent washing, beads with captured transcripts were eluted and cDNA synthesis was performed at 37°C for 20 min with shaking, followed by exonuclease I treatment at 37°C for 30 min and heat inactivation at 80°C according to the manufacturer's instructions using the BD RhapsodyTM cDNA Kit (Cat. No. 633773).

Finally, mRNA libraries were prepared according to the manufacturers protocol (mRNA Targeted Library Preparation with the BD RhapsodyTM and BD RhapsodyTM Express Single-Cell Analysis Systems (Doc ID: 210968) using the BD RhapsodyTM Targeted mRNA and AbSeq Amplification Kit (Cat. No. 633774) in combination with the custom primer libraries. These custom primers for 146 transcripts corresponding to 78 genes in the 6MB genomic space (+/− 3MB from the PHOX2B TSS) on chr4 and a probe for capturing gRNAs were designed by BD. The target enrichment is achieved using two PCR reactions with two different sets of transcript specific PCR primers. The probes for gRNA capture were added as a supplemental panel in both nested PCR reactions. Upon addition of indecis, the libraries were quantified using a Qubit Fluorometer with the Qubit dsDNA HS Assay Kit (Thermo Fisher Scientific, #Q32851) and size-distribution was analyzed using the Agilent DNA High Sensitivity Kit (Agilent Technologies, #5067 − 4626) on a TapeStation 4200 system. Paired-end sequencing (2x75nt) was performed on a NextSeq 500/550 using a HighOutput v2 Kit for 150 cycles with a 20% PhiX spike-in.

### Computational analysis of TESLA-seq data

The raw sequencing output files were preprocessed using BD's Rhapsody Targeted Analysis pipeline (https://scomix.bd.com/hc/en-us/articles/360019763251-Bioinformatics-Guides) on the Seven Bridges Platform (https://www.sevenbridges.com/bdgenomics/), which generated a cell-by-gene count matrix corrected for base-calling errors in cell barcodes, followed by quality-filtering on cells. Processed count matrices of two sample runs contain 12,597 (run1) and 19,406 (run2) cells, respectively, with features consisting of 1,046 guide RNAs (gRNA) targeting the 2Mb region around the PHOX2B gene, 52 control gRNAs, and 78 genes within the 6Mb region around the PHOX2B gene. The count matrices were loaded into the R environment for further preprocessing with Seurat v4.[82] Logarithm base 10 was applied to both the gene count matrices and gRNA count matrices. The gRNA count values display a clear bimodal pattern, suggesting that the peak of lower values could be noise. After setting the lower values in gRNA matrices to zero, gRNAs with no count values across cells and genes with no expression were removed. Cells with either low gRNA values or low gene expression were filtered. After filtering, run1 has 6,196 cells with 1,090 gRNAs and 62 genes, while run2 has 10,363 cells with 1,097 gRNAs and 64 genes.

The quality-filtered matrices of two runs were normalized using SCTransform[83] and integrated with batch correction done by linear regression between runs (SCTransform argument 'var.to.regress'). Cells that had less than two normalized and batch-corrected guide counts were removed and genes that have no expression following the cell removal were filtered. After filtering, run1 has 6,188 cells with 1,086 gRNAs and 59 genes, while run2 has 10,347 cells with 1,086 gRNAs and 59 genes. Of note, we have also tested 'LogNormalize' method from Seurat for normalization of our data and obtained similar results. We opted for SCTransform since in addition to normalizing for sequencing depth, it further accounts for technical variation, performs variance stabilization, and includes feature selection. Compared to classical normalization methods, SCTransform model mitigates technical noise more effectively while preserving biological variation.

To test the effect of each gRNA and avoid potential confounding caused by nearby gRNAs, differential expression analysis of each targeted gene was performed between cells that contain the considered gRNA and all other cells that contain other gRNAs targeting at least 400 kb away from the considered gRNA. MAST.cov test was chosen to perform differential expression analysis as recommended for targeted single-cell data[17] with the number of genes captured per cell treated as the covariate. A significance level of 0.05 adjusted $p$-value was used to identify regulation pairs of gRNA/CaRE and gene. Plots were generated using R packages ggplot2 v3.3.2[84] and Sushi v1.7.1.[85]

### Nuclear capture C

nuCapture C was performed according to[46] in SH-SY5Y cell line. Single-stranded DNA probes for genes in the 6MB space surrounding PHOX2B were obtained from IDT as an xGen Lockdown Pool and are listed in Table S2. Paired-end sequencing (2x150nt) was performed on a NextSeq 500/550 using a HighOutput v2 Kit for 300 cycles (Illumina #FC-420-1004, discontinued).

### Nuclear capture C data processing

Nuclear Capture C reads were initially processed using the HiCUP pipeline[86] with Bowtie2 mapping to hg38. Mapped reads were then converted to.chinput format using chicagoTools and used as input to chicagoPipeline[87] along with DpnII genomic fragment regions targeted for capture during library preparation.

### Hi-C data processing

Raw reads from published IMR5/75 cells Hi-C experiments[88] were downloaded from SRA (PRJNA622577). Processing steps were implemented within the Snakemake framework.[89] Hi-C reads were initially processed using the Juicer pipeline.[90] TADs were called using the insulation method[91] with default settings (–is500000–nt0–ids250000–ss0–immean) after dumping valid interactions at 25 kb binned resolution using juicer-tools and converting formats using HiTC.[92] For scoring Hi-C interactions, valid interactions were dumped by juicer-tools at fragment resolution, filtered to remove interactions less than 20 kb apart, format converted using custom scripts, and then subjected to shuffling and scoring using the SHAMAN method.[93]

### Random forest model for predicting significant CaRE-gene pairs from genomic features

Random forest models were trained with scikit-learn tools v1.2.1[94] in Python. The model was trained to learn how well genomic features, linear distance between a perturbed region and gene, CaptureC score, ATAC signal, and epigenetic markers can predict the regulatory relationships between CaREs and genes. The top 50 and bottom 50 regulatory pairs, ranked by adjusted $p$-values, were selected as the target of prediction (1 for top 50, 0 for bottom 50), and the genomic features were treated as the explanatory features.

### TESLA-seq multimodal integration analysis

Nuclear capture-c scores were assigned to CaRE-gene pairs by intersecting 25kb regions surrounding CaRE midpoints and gene TSSs with fragment-target pair coordinates and scores as output by the Chicago pipeline using bedtools pairtopair (-slop 12500;[70]) and taking the maximum Chicago score per pair within this 25kb interaction square. Pairs were classified as interacting if this maximum Chicago score was greater than 3.

Hi-C scores were assigned to CaRE-gene pairs by taking the highest Shaman score within a 25kb square centered on the CaRE midpoint and gene TSS using Shaman scripts. A CaRE-gene pair was classified as Hi-C interacting if the highest Shaman score in this 25kb square was greater than or equal to 25.

Epimap[42] link coordinates and scores were downloaded by tissue group and concatenated. Epimap region coordinates were shifted to midpoints and gene TSS coordinates were assigned via Ensembl gene ID. These pair coordinates were intersected with CaRE-midpoint-TSS pairs using bedtools pairtopair (-slop 2500) and the maximum score within this 5kb square was assigned to CaRE-gene pairs.

Abc file AllPredictions.AvgHiC.ABC0.015.minus150.ForABCPaperV3.txt was downloaded, converted to bed format with region coordinates, gene name, and ABC.score, lifted over to hg38, target gene TSS coordinates retrieved via gene name, and the maximum ABC.score across tissues was taken for each region-geneTSS pair. Regions coordinates were then shifted to midpoints and these pairs were intersected with CaRE-midpoint-geneTSS coordinates using bedtools pairtopair (-slop 2500) and the maximum ABC.score in this 5kb square was assigned to each CaRE-gene pair.

Histone modification ChIP-seq and ATAC-seq signal was assigned to CaREs and gene TSSs using 1kb windows centered on CaRE midpoints or gene TSSs and summarizing the signal as described above in section "analysis of Ca/iRE features in neuroblastoma data".

Chromatin RNA-seq signal was assigned to genes as the mean counts between two replicates as output by STAR.

In order to classify CaREs by previous evidence, we used our own ATAC-seq and published ChIP-seq from neuroblastoma cells (except for H3K27me3), called peaks as described above, and concatenated and merged all resulting regions as neuroblastoma regulatory evidence. Epimap DHS, dyad, promoter, and enhancer regions for hg38 were downloaded from/www.meuleman.org/and personal.broadinstitute.org/cboix/epimap/mark_matrices/and DHS coordinates were matched to Epimap annotations using identifiers. The resulting file and the merged nueroblastoma regulatory evidence were each separately used to find the closest element to each CaRE using bedtools closest (-d -t "first";[70]) and a true assignment was made if the distance of the closest element was less than or equal to 250bp. Dyads were considered promoters. If a CaRE was within 250bp of a neuroplastoma reulatory evidence element, it was classified as "neuroblastoma" ("nb"), otherwise if it was close to an Epimap element, it was classified as "other", otherwise it was classified as "no evidence found" ("noEF").

### Supplemental note 1 - CRISPRi screen

In addition to the CRISPRa phenotypic screen, we performed an initial orthogonal phenotypic CRISPRi screen with the dCas9-KRAB construct and the same gRNAs. Although this construct has been widely adopted, dCas9–KRAB is now known to suffer from inefficient knockdown and poor performance. Specifically, recent extensive benchmarks on large panels of CRISPRi constructs demonstrated a comparatively poor performance for dCas9-KRAB compared to recent alternatives.[20,21] The performance of dCas9–KRAB is even less effective on non-coding elements. Of note, since different CRISPRi constructs can have different efficacies in different cell lines, it is also possible that dCas9–KRAB is not an efficient repressor in the SH-SY5Y cell line that we used in the study.

We extensively compared the CRISPRa and CRISPRi phenotypic screens (Figure S6). We first demonstrated that we can repress PHOX2B using dCas9-KRAB and that this causes a reduction in cellular viability (Figure S6A,B). The data from the CRISPRi screen is notably noisier and sparser than our CRISPRa data, in line with the evidence of poor performance for dCas9-KRAB compared to recent alternatives (Figure S6C-G;[20,21]). The data thus does not allow us to comprehensively define regulatory elements based on neighboring gRNAs behaving in the same manner - CRISPRi responsive elements (CiREs), as we did to define CaREs. Once we analyze the data using regions defined by the CRISPRa experiments - CaREs, we find that some of the individual CiRE hits appear to be true positives (Figure S6E-G).

### Supplemental note 2 - Characteristics of CRISPR-mediated activation

Although available evidence points to dCas9-VPR effects being precise and localized without a possibility of epigenetic spreading as in the case of histone modifiers,[28] we can not completely rule out that regulatory elements in close 3D proximity to an artificially activated element may become activated due to a local increase in the transcriptional machinery concentration. To address the question of the spreading of the effect of CRISPRa in our data, we examined pairs of closest neighboring CaREs in the linear genome, where both have significant target genes as hits (Figure S7A). Although some of the CaREs share target genes (which can be expected from

neighboring REs that regulate the same gene), none of the pairs - except for one pair both targeting PHOX2B - share the same set of target genes, which would be expected if the CRISPRa effect was spreading.

Additionally, to address whether CRISPRa binding interferes intragenically with RNA polymerase, we have examined the effect of our CaREs located within gene bodies on the expression of those genes. We do not find evidence of RNA polymerase blocking with this analysis (Figure S7B).

Due to the time period elapsing between transduction and single-cell readout, we cannot rule out additional secondary/indirect regulatory effects. While further studies are needed to confidently distinguish between the two, the observed evidence for prior 3D interaction between significant CaRE-gene pairs as well as the overlap with known regulatory elements and targets suggest that we are primarily identifying direct effects. To further address this question in the context of our data, which focuses on a specific locus, we have used available PHOX2B ChIP-seq data to determine PHOX2B binding sites within the significant CaREs defined by TESLA-seq. We then took their four target genes and compared expression of the cells in which the PHOX2B promoter was perturbed to the cells in which the PHOX2B promoter was not perturbed (Figure S7C). We do not find obvious differences between the expression of these target genes in cells in which PHOX2B is activated, in line with a lack of downstream secondary effects.

## QUANTIFICATION AND STATISTICAL ANALYSIS

Details on statistical tests can be found in the Results, figure legends, and Method details sections.

