## [Document S2. Transparent peer review records for Vučićević et al · Cell Genomics]

Sensitive dissection of a genomic regulatory landscape using bulk and targeted single-cell activation

Dubravka Vučićević, Che-Wei Hsu, Lorena Sofia Lopez Zepeda, Martin Burkert, Antje Hirsekorn, Ilija Bilić, Nicolai Kastelić, Markus Landthaler, Scott Allen Lacadie, Uwe Ohler

Summary

Initial submission: Received : Nov 21, 2024

Scientific editor: Sara Rohban

First round of review: Number of reviewers: 2
Revision invited : Jan 03, 2025
Revision received : May 09, 2025

Second round of review: Number of reviewers: 2
Accepted : Aug 4, 2025

Data freely available: YES

Code freely available: YES

This transparent peer review record is not systematically proofread, type-set, or edited. Special characters, formatting, and equations may fail to render properly. Standard procedural text within the editor's letters has been deleted for the sake of brevity, but all official correspondence specific to the manuscript has been preserved.

Referees' reports, first round of review

Reviewer #1:

Functional characterization of cis-regulatory elements (CREs) and their target genes remains a critical challenge in the field. To overcome this challenge, the authors have developed TESLA-seq, a single-cell approach that integrates pooled CRISPRa perturbation with targeted single-cell RNA-seq (using a microwell-based method) for high-throughput mapping of the impact of CRE activation on the expression of nearby genes. This approach enables the identification of novel CRE-gene linkages that were not captured in previous studies, revealing highly complex regulatory networks between CREs and their associated genes. Overall, this is an exciting approach with broad applications for the systematic functional analysis of non-coding regions. Below are a couple of comments that should be addressed before publication:

1. The authors performed both CRISPRi and CRISPRa screens for the same regions, but there is minimal analysis comparing the results. A comparative analysis would strengthen the manuscript by highlighting the complementary insights gained from each approach.
2. One potential concern with the CRISPRa/i screening is the possible spread of epigenetic markers, which may introduce non-specific effects in nearby regions. This issue is not evaluated in the manuscript. Additionally, does CRISPRa binding interfere with RNA polymerase transcription when binding to intragenic regions?
3. Another potential concern is the normalization method for targeted profiling compared to whole-transcriptome profiling, which typically uses total counts as a normalization factor. How does the chosen normalization strategy affect the results?
4. The CRE-gene linkages identified by the authors could function in cis- or trans-manner (e.g., the alteration of one gene leading to changes in other genes). How do the authors distinguish between these two effects?
5. For broader application of the approach, it would be helpful if the authors could share a detailed step-by-step experimental protocol. Additionally, plasmids should be deposited in Addgene prior to publication to facilitate replication and further research by the scientific community.

Reviewer #2:

The paper describes TESLA-seq, an approach for identifying CRISPRa-responsive elements of genes through a two-pronged technique: (1) a tiling fitness screen, (2) subsequent targeted single-cell RNA-seq-based profiling to link hits to the genes that they regulate. The primary advance on the conceptually similar TAP-seq approach is the use of CRISPRa and a commercial single-cell RNA sequencing platform that does targeted enrichment as part of its workflow. I think the primary weaknesses of the paper in its current form are: (1) Some of the quantitative modeling is poorly described, leaving some questions in my mind about the strength of the connections being made and whether there are false positive issues. (2) As a methods-focused paper, I think more effort should be made to identify the quantitative limitations on resolution. A final consideration that I think could enhance the paper is that a matched CRISPRi screen was performed but is not analyzed. It seems that this could help support the novelty of the CRISPRa approach here (given that a number of papers using CRISPRi to target enhancers with various readouts exist). I do think these issues are addressable through further computational analyses, after which the paper would be suitable for publication in Cell Genomics.

I'm confused how 758 negative hits and 27 positive hits at guide level get turned into 546 negative and 83 positive CaREs. From reading the methods, it appears that guide significance was not taken into account in defining CaREs? Though I understand that fusing measurements is expected to increase power, it's unclear to me how many measurements are ultimately supporting the definition of each CaRE. The data in 1H look like other tiling screens: i.e., either quite noisy, or alternatively quite finely structured, depending on your level of optimism. Is there an argument for why the more complicated modeling approach was needed over something simple like a Mann-Whitney test? Side note: modeling uses somewhat non-standard notation. What does "[1] is the overall estimate for Time" mean? I assume this is something along the lines of a constant representation over time for each guide?

You state that you identify CaREs in many promoters that led to changes in growth. How consistent are these effects? I.e. do there exist positive and negative CaREs within promoters? I'm wondering because the hit rate appears to be high. Most gene-level CRISPRa fitness screens have found relatively few hits compared to other modalities.

The predictive modeling of CaREs from chromatin features is not explained clearly. The methods section is light on details, as is the explanation for why particular TFs were included. The statement "could predict significant CaREs" really needs a quantitative measure attached. How good are the predictions? How is the train/test split being done here?

Given that the CRISPRi screen was performed, it seems worthwhile to at least include some comparisons to it. I think this would significantly enhance the argument for novelty as it would show why CRISPRa reveals different connections from CRISPRi.

Though I realize this is covered in more detail with the TESLA-seq data, I do think some similar analysis in terms of comparisons to other datasets could be provided for the fitness-defined CaREs. I'd be interested to know if the CaREs that intersect more classical features (like chromatin accessibility or TF binding sites are stronger hits) behave differently in some way or are stronger hits in general. I'm frankly a bit perplexed by the statement "the hits are equally distributed across the whole 2 Mb space" (which is not supported by any quantitative metric or figure). This seems unintuitive to me, as I guess I'd expect to see some proximity bias. A more direct comparison to TAP-seq, if possible, would be helpful for people considering the two techniques.

There is a typo in 2A "CRSPRa" should be "CRISPRa". There is also a typo on line 213.

Regarding 2H, it would be helpful to annotate the mean expression level here. Similarly 2E I think could include all targeted TSSes and plot as a function of expression level. Clearly the expression level will affect which links can reliably be established.

When you talk about skipping genes, I assume you are counting non-coding transcripts as genes? Perhaps worth breaking this out as coding vs. non-coding. Are the skipped genes also just very low expression? That could explain why there is no obvious effect.

Regarding 3G, training a random forest classifier on 100 examples will probably result in strong overfitting. Why not just use logistic regression or even simply compare summary statistics for the different features, given that you are preselecting the strongest examples in each class?

In 4A, are all of the depicted genes reliably captured? I'm wondering again if expression level partly explains why some appear to have no regulators.

There is an overall question of what is going on with the other 90% of fitness-defined CaREs, as only 60 CaREs participated in TESLA-seq interactions out of >600 identified in the original screen. Though the paper favors an interpretation of conservative selection, I do think two other alternatives are: (1) off-target effects of CRISPRa and (2) false positives from an overly permissive pipeline for the fitness data. I so think some consideration should be given to these possibilities.

A final point worth considering is whether the tiling fitness screen will ultimately be necessary. Various compressed experimental designs for studying enhancers have been put forward, including the Gasperini et al. paper. It seems possible that these could be used alongside the single-cell readout here to do tiling directly.

Authors' response to the first round of review

Reviewer #1: Functional characterization of cis-regulatory elements (CREs) and their target genes remains a critical challenge in the field. To overcome this challenge, the authors have developed TESLA-seq, a single-cell approach that integrates pooled CRISPRa perturbation with targeted single-cell RNA-seq (using a microwell-based method) for highthroughput mapping of the impact of CRE activation on the expression of nearby genes. This approach enables the identification of novel CRE-gene linkages that were not captured in previous studies, revealing highly complex regulatory networks between CREs and their associated genes. Overall, this is an exciting approach with broad applications for the systematic functional analysis of non-coding regions. Below are a couple of comments that should be addressed before publication:

1. The authors performed both CRISPRi and CRISPRa screens for the same regions, but there is minimal analysis comparing the results. A comparative analysis would strengthen the manuscript by highlighting the complementary insights gained from each approach.

For our CRISPRi experiments we used dCas9-KRAB. Although this construct has been widely adopted, the dCas9-KRAB system is now known to suffer from inefficient knockdown and poor performance. Specifically, recent extensive benchmarks on large panels of CRISPRi constructs demonstrated a comparatively poor performance for dCas9-KRAB compared to recent alternatives (Alerasool et al., 2020, Nuñez et al., 2021). The performance of dCas9-KRAB is even less effective on non-coding elements.

Below is a comparative analysis between the CRISPRa and CRISPRi screen (Figure R1E-G). The data from the CRISPRi screen is notably noisier and sparser than our CRISPRa data in line with the evidence of poor performance for dCas9-KRAB compared to recent alternatives (Alerasool et al., 2020, Nuñez et al., 2021). The data thus does not allow us to comprehensively define regulatory elements based on neighboring gRNAs behaving in the same manner - CRISPRi responsive elements (CiREs), as we did to define CaREs. Once we analyze the data using regions defined by the CRISPRa experiments - CaREs, we find that some of the individual CiRE hits appear to be true positives. We include the CRISPRi data and the comparison between the CRISPRa and CRISPRi phenotypic screen in Supplemental Note 1 so as not to distract from the main story of our manuscript.

Figure SN1. CRISPRi phenotypic screen. **A)** PHOX2B expression changes upon repression with dCas9-KRAB with control or gRNAs targeting the promoter of Phox2B in SH-SY5Y cell line. Relative expression of target genes is determined by RT- qPCR and normalised to GAPDH. Shown are fold changes relative to the control. Expression of actin is shown as a control. **B)** MTT viability assay with control or gRNAs targeting the promoter of Phox2B in a SH-SY5Y-KRAB (CRISPRi) cell line. **C)** Volcano

plot showing the log-fold change of gRNA representation between the first and last time-point of the CRISPRi experiment. Significantly enriched/depleted gRNAs (FDR<0.05) are shown in red and blue, respectively. **D)** Volcano plot showing the results for our CRISPRi screen, each dot corresponds to a CaRE defined by the CRISPRa screen. The x axis shows the slope calculated by MLM. Significantly enriched/depleted CaREs (FDR<0.05) are shown in red and blue, respectively. **E)** CiREs signal around the PHOX2B locus (+/- 1 MB). From bottom to top: annotation for PHOX2B and its location within the genome, ChIP-seq signal for HAND2, PHOX2B, H3K27ac, and ATAC-seq signal in SH-SY5Y cell line. At the top is the score and direction (blue for depletion, red for enrichment) of CiREs. *P*-values shown are adjusted for multiple testing (FDR). **F-G)** Comparison between CRISPRa and CRISPRi screen at the gRNA (F) and CaRE/CiRE level (G) in adjusted p-value (left) or fold change (estimate, right).

2. One potential concern with the CRISPRa/i screening is the possible spread of epigenetic markers, which may introduce non-specific effects in nearby regions. This issue is not evaluated in the manuscript. Additionally, does CRISPRa binding interfere with RNA polymerase transcription when binding to intragenic regions?

Spreading has mostly been described in the literature in the context of gene silencing through DNA methylation, H3K9me3-marked heterochromatin, and the polycomb mark H3K27me3. We are unaware of examples where lateral spreading was described for transcriptionally permissive or active chromatin states. In the case of the CRISPRa construct we used, dCas9-VPR, spreading is unlikely since it is actually not an epigenetic modifier: It consists of three potent transcriptional activators: VP64, p65, and Rta, which act synergistically to recruit the host cell's transcriptional machinery and strongly enhance gene expression (Chavez 2015). Accordingly, studies addressing the off-target effects of dCas9-VPR demonstrated that they are very low or undetectable (Chavez 2016; Schoger 2019; Bohm 2020), and in Chavez 2016, activation of one gene followed by RNA-seq could only detect that one gene was differentially expressed. Furthermore, ChIP experiment on dCas9-VPR (Schoger 2019) demonstrates the precise and specific binding at the intended location. We point this out in the result section: We tested several CRISPRa constructs for robust activation of known enhancers (Chavez et al., 2016) (Figure S1A) and selected dCas9-VPR for further experiments due to having the strongest activation and since it consists of three potent transcriptional activators that are shown to be precise and localized (Chavez et al., 2016).

While highly suggestive, these points do not completely rule out the possibility that regulatory elements in close 3D proximity to an artificially activated element could not also become activated due to the local increase in the transcriptional machinery concentration. To address the question of the spreading of the effect of CRISPRa in our TESLA-seq data, we examined pairs of closest neighboring CaREs in the linear genome, where both have significant target genes as hits (cf table below, Table R1). Although some of the CaREs share target genes (which can be expected from neighboring REs that regulate the same gene), none of the pairs - except for one pair both targeting PHOX2B - share the same full set of target genes, which would be expected if the CRISPRa effect was spreading.

CaRE	Closest CaRE	distance (bp)	CaRE_target	closestCaRE_target
CaRE_1	CaRE_3	3671	RBM47,NSUN7,RP S7P7	PDS5A,TLR1
CaRE_171	CaRE_174	9631	NDUFB4P12	APBB2
CaRE_174	CaRE_179	9129	APBB2	TLR1
CaRE_553	CaRE_557	4752	PHOX2B,LINC0068 2	PHOX2B
CaRE_561	CaRE_562	1557	PHOX2B	PHOX2B
CaRE_668	CaRE_669	1133	CHRNA9	BEND4
CaRE_671	CaRE_669	2254	BEND4,AC098590. 1	BEND4
CaRE_729	CaRE_730	1020	SHISA3,ZBTB12BP	BEND4,SHISA3
CaRE_740	CaRE_741	826	SHISA3,BEND4	SHISA3
CaRE_789	CaRE_790	654	SHISA3	SHISA3,TLR6,UGDH
CaRE_803	CaRE_806	5134	SHISA3	SHISA3,TMEM156,PABP C1P1
CaRE_828	CaRE_829	1266	SHISA3,TLR6,LINC 02265	SHISA3
CaRE_836	CaRE_831	7172	ATP8A1	SHISA3

Table R1. Table showing the closest significant CaRE neighbors and their target genes identified by TESLA-seq.

To address whether CRISPRa binding interferes intragenetically with RNA polymerase, we have examined the effect of our CaREs located within gene bodies on the expression of those genes. We do not find evidence of RNA polymerase blocking with this analysis (Figure R2).

Figure R2. Genes expression comparison in cells in which the following CaREs are targeted: CaREs targeting the promoter of the gene depicted (promoter proximal), CaREs within genes that are not at the promoters of genes (intragenic nonproximal) and CaREs that are targeting genes other than the one depicted (control).

These are interesting questions, they are limited by the number of targets within this locus and we feel beyond the main scope of the study. We include this analysis in the Supplementary Note 2.

3. Another potential concern is the normalization method for targeted profiling compared to whole-transcriptome profiling, which typically uses total counts as a normalization factor. How does the chosen normalization strategy affect the results?

We assume the reviewer is referring to the ‘LogNormalize’ method in Seurat, which is a widely used classical normalization approach. This method normalizes each cell by dividing its gene expression values by the total counts, scaling by a factor of 10,000, and applying a natural log transformation. It effectively accounts for differences in sequencing depth and is commonly used in single-cell RNA-seq analysis.

In contrast, we opted for SCTransform, which employs a regularized negative binomial regression model for normalization. While SCTransform also adjusts for sequencing depth, it further accounts for technical variation, performs variance stabilization, and includes feature selection. Compared to classical normalization methods, SCTransform models and mitigates technical noise more effectively while preserving biological variation. This leads to improved detection of highly variable genes, reduced batch effects, and enhanced performance in downstream clustering and dimensionality reduction. For our targeted profiling data, both LogNormalize and SCTransform produced similar distributions of total normalized values per cell across replicates (Figure R3). Additionally, for gRNA calling, we found that the choice of normalization strategy for gRNA counts did not affect the results. These findings suggest that

our normalization approach does not introduce significant biases and is unlikely to substantially impact downstream analyses. We included the following statement in the methods section:” Of note, we have also tested the

‘LogNormalize’ method from Seurat for normalization of our data and obtained similar results. We opted for SCTransform since in addition to normalizing for sequencing depth, it further accounts for technical variation, performs variance stabilization, and includes feature selection. Compared to classical normalization methods, SCTransform model mitigates technical noise more effectively while preserving biological variation.”

Figure R3. Comparison of normalization methods.

- A) Distribution of raw UMI counts per cell.
- B) Distribution of normalized values per cell (LogNormalize)
- C) Distribution of normalized values per cell (SCTransform)
- D) Boxplot of gRNA per cell after filtering with threshold of 1 or 2 counts/units (gRNA calling) on raw counts or normalized values

4. The CRE-gene linkages identified by the authors could function in cis- or trans-manner (e.g., the alteration of one gene leading to changes in other genes). How do the authors distinguish between these two effects?

This is a common concern with large-scale perturbation experiments. To minimize these secondary effects, cells were collected as soon as the antibiotic selection for the integrated gRNAs was completed. However, due to the time period elapsing between transduction and single-cell readout, we cannot rule out additional secondary/indirect regulatory effects. While further studies are needed to confidently distinguish between the two, the observed evidence for prior 3D interaction between significant CaRE-gene pairs (Figure 3F right, Figure 4A) as well as the overlap with known regulatory elements and targets (Figure 4A, Figure S4A) suggest that we are primarily identifying direct effects. To further address this question in the context of our data, which focuses on a specific locus, we have used available PHOX2B ChIP-seq data to determine PHOX2B binding sites within the significant CaREs defined by TESLA-seq. We then took their four target genes as identified by TESLA-seq and compared expression of the cells in which the PHOX2B promoter was perturbed to the cells in which the PHOX2B promoter was not perturbed (Figure R4). We do not find obvious differences between the expression of these target genes in cells in which PHOX2B is activated, therefore suggesting that we are not detecting downstream secondary effects. We have included this analysis in the Supplementary note 2.

We have elaborated on this in the discussion: “ However, due to the time period elapsing between transduction and single-cell readout, we cannot rule out additional secondary/indirect regulatory effects. While further studies are needed to confidently distinguish between the two, the observed evidence for prior 3D interaction between significant CaRE-gene pairs (Figure 3F right, Figure 4A) as well as the overlap with known regulatory elements and targets

(Figure 4A, Figure S4A) suggest that we are primarily identifying direct effects.”

Figure R4. Gene expression comparison of the genes targeted by CaREs that have PHOX2B binding sites in cells in which the PHOX2B promoter was activated (blue) and cells in which PHOX2B promoter was not activated (red).

5. For broader application of the approach, it would be helpful if the authors could share a detailed step-by-step experimental protocol. Additionally, plasmids should be deposited in Addgene prior to publication to facilitate replication and further research by the scientific community.

Thank you for the suggestion. A detailed experimental protocol has been included in the TESLA-seq methods section and the plasmids have been deposited in Addgene (#233066 and #233067).

Reviewer #2: The paper describes TESLA-seq, an approach for identifying CRISPRa-responsive elements of genes through a two-pronged technique: (1) a tiling fitness screen, (2) subsequent targeted single-cell RNA-seq-based profiling to link hits to the genes that they regulate. The primary advance on the conceptually similar TAP-seq approach is the use of CRISPRa and a commercial single-cell RNA sequencing platform that does targeted enrichment as part of its workflow. I think the primary weaknesses of the paper in its current form are: (1) Some of the quantitative modeling is poorly described, leaving some questions in my mind about the strength of the connections being made and whether there are false positive issues. (2) As a methods-focused paper, I think more effort should be made to identify the quantitative limitations on resolution. A final consideration that I think could enhance the paper is that a matched CRISPRi screen was performed but is not analyzed. It seems that this could help support the novelty of the CRISPRa approach here (given that a number of papers using CRISPRi to target enhancers with various readouts exist). I do think these issues are addressable through further computational analyses, after which the paper would be suitable for publication in Cell Genomics.

I'm confused how 758 negative hits and 27 positive hits at guide level get turned into 546 negative and 83 positive CaREs. From reading the methods, it appears that guide significance was not taken into account in defining CaREs? Though I understand that fusing measurements is expected to increase power, it's unclear to me how many measurements are ultimately supporting the definition of each CaRE. The data in 1H look like other tiling screens: i.e., either quite noisy, or alternatively quite finely structured, depending on your level of optimism. Is there an argument for why the more complicated modeling approach was needed over something simple like a Mann-Whitney test? Side note: modeling uses somewhat non-standard notation. What does "[1] is the overall estimate for Time" mean? I assume this is something along the lines of a constant representation over time for each guide?

Effects on the gRNA level and the CaREs level are based on different models – while the gRNA analysis is included to allow for an immediate assessment compared to previous studies, the main computational novelty is the analysis of multiple guides over time within a linear mixed model, which we use to define CaREs. We apologize that this was unclear, and we have rewritten the methods section to clarify the analysis of the phenotypic screen.

Although we designed our analyses to be solid and stringent, the high hit rate for the fitness screen is indeed surprising. Phox2B may be part of a larger genomic domain containing several genes important for neuroblastoma cell growth. To clarify the contribution of how many measurements are supporting the definition of each CaRE we are including in the manuscript the plots in Figure R5 in Figure 1G.

Figure R5 Number of gRNAs that contribute to each CaRE stratified by the width of the CaREs.

You state that you identify CaREs in many promoters that led to changes in growth. How consistent are these effects? I.e. do there exist positive and negative CaREs within promoters? I'm wondering because the hit rate appears to be high. Most gene-level CRISPRa fitness screens have found relatively few hits compared to other modalities.

We are unfortunately not entirely sure here what the reviewer is referring to. If it is about CaREs within promoters that have an effect on the phenotype, we find 19 significant promoter CaRE hits, and all lead to a reduction in cellular growth/viability. In the case of PHOX2B, this is something we expect from the literature since both activation and repression of this gene lead to cellular death.

We point this out in the Result section of the phenotypic screen : “Besides PHOX2B, the promoters of several other genes coincide with CaREs that led to a reduction of cellular growth and survival: RP11-227F19 (non-coding RNA that shares a promoter with PHOX2B), AC105389, APBB2, ATP8A1 and AC096734 that share a promoter, BEND4, DCAF4L1, NSUN7, AC105389, AC108210, AC024022, SHISA3, SLC30A9, TMEM33, UCHL1 and UCHL1.AS1 that share the promoter, Y_RNA and AC105389.1. Some of them are known to affect cellular growth/proliferation: SHISA3 is a known tumor suppressor (Chen et al., 2014, Shahzad et al., 2020, Zhang et al., 2024); APBB2 plays a role in Alzheimer's disease (Li et al., 2005, Golanska et al., 2013) and cell cycle (Bruni et al., 2002, Zhou et al., 2021); UCHL1 promotes cellular proliferation in cancer (Kwan et al., 2020, Mondal et al., 2022).”

Additionally, we highlight these in the TESLA-seq result section: “Notably, most of the TESLA-assayed CaRE target genes whose promoters were hits in the bulk growth screen are targets in the TESLA-seq, and these interactions are especially strong and significant (Figure 2H - highlighted genes, Figure 2J). This is expected since the TESLA-seq guides were chosen based on their phenotypic effect in the tiling bulk screen (Figure 1), and these results can, therefore be considered as validation for CaREs affecting growth/viability. “

The predictive modeling of CaREs from chromatin features is not explained clearly. The methods section is light on details, as is the explanation for why particular TFs were included. The statement "could predict significant CaREs" really needs a quantitative measure attached. How good are the predictions? How is the train/test split being done here?

Our selection of TFs is based on previously described "core regulators" in neuroblastoma, so we included all available neuroblastoma data (Henrich et al., 2016, Durbin et al., 2018, Wang et al., 2019, Boeva et al., 2017, Zhang et al., 2020). We have now added a model evaluation based on ROC-AUC (Figure R6), but we want to point out that our main aim is to judge feature importance using the trained model and do not deem predictive performance of such a model key for our purposes. We have expanded the methods section to clarify the details of the model. We have also expanded and modified the results section accordingly with: "logistic regression models (see methods) predicted CaREs based on accessible chromatin, and histone modifications H3K27ac and H3K4me1 (Figure 1J) significantly better than random baseline models (Figure S1D, mean ROC-AUC 0.60). Further, key neuronal transcription factors MYCN, HAND2, ISL1, ASCL1, GATA3 and GATA2 emerged as significant predictors when incorporated into the model (Figure 1K)".

Figure R6. Evaluation of epigenetic model for CaREs prediction. Left panel: Receiver operator characteristic (ROC) of trained CaRE models (light orange, $n=50$) and random models (light blue, $n=50$) from 10 repeats of a 5-fold cross validation each. Mean ROC curve for CaRE and random models in dark orange and blue respectively. TPR: True positive rate, FPR: False positive rate. Right panel: Distributions of ROC area under the curve (ROC-AUC) for CaRE and random models. Boxplot midlines mark median; upper and lower hinges extend to first and third quartile; upper and lower whiskers extend to the smallest and largest value $\max. 1.5 \times \text{IQR}$. Comparison p-value from two-sided Wilcoxon ranksum test.

Given that the CRISPRi screen was performed, it seems worthwhile to at least include some comparisons to it. I think this would significantly enhance the argument for novelty as it would show why CRISPRa reveals different connections from CRISPRi.

Please refer to Reviewer 1 comment 1. To address this comment we have included a Supplementary note 1 in which we compared the CRISPRa and CRISPRi screen.

Though I realize this is covered in more detail with the TESLA-seq data, I do think some similar analysis in terms of comparisons to other datasets could be provided for the fitness-defined CaREs. I'd be interested to know if the CaREs that intersect more classical features (like chromatin accessibility or TF binding sites are stronger hits) behave differently in some way or are stronger hits in general. I'm frankly a bit perplexed by the statement "the hits are equally distributed across the whole 2 Mb space" (which is not supported by any quantitative metric or figure). This seems unintuitive to me, as I guess I'd expect to see some proximity bias.

We have included a more detailed analysis of the fitness-defined CaREs. As for the TESLA-seq defined CaREs, we have addressed their direct association to open chromatin and active histone marks (H3K4me1, H3K4me3, H3K27ac) in Figure S1E-H. We have also included in the supplement the analysis of their association to previously defined REs in EpiMap data in Figure S1I-K (Figure R7). The following sentence was also included in the manuscript: "We compared CaREs with evidence in neuroblastoma to those with evidence in other cell lines as described in the EpiMap database and found that the ones with evidence in neuroblastoma are more significant than the ones without evidence in neuroblastoma (Figure S1I-K, Boix et al., 2021)".

We have clarified the statement that "the hits are equally distributed across the whole 2 Mb space"; this was indeed too colloquial and unclear, we referred to the presence of hits outside of the super enhancer region in the proximity to the Phox2B gene. : "The highest scoring CaREs are in the proximity of the PHOX2B promoter. They overlap a super-enhancer predicted to regulate PHOX2B expression in neuroblastoma ((Boeva et al., 2017); Figure 1I).".

A)

B)

Figure R7. Characteristics of CaREs that affect cellular phenotype. A) From top to bottom: the association of CaREs to open chromatin -ATAC, H3K4me3 signal, H3K4me1 signal and H3K27ac signal relative to the CaRE fold change (estimate, left) or adjusted p-value (right). B) Number of CaREs in each category,

nb, other, noEF grouped by the EpiMap classification indicated with colours (left). Comparison between different CaRE evidence categories (nb, other, noEF) in CaRE fold change (estimate, middle) and adj. P-value (right).

A more direct comparison to TAP-seq, if possible, would be helpful for people considering the two techniques.

While TESLA-seq and TAP-seq share some similarities, materials and experimental conditions differ and they cannot be directly compared in an apples-to-apples manner. Based on our evaluation, our data consistently show a capture efficiency of 0.99, regardless of whether the counts are normalized or whether a stricter or more lenient filtering threshold is applied for gRNA calling. This efficiency remains consistent across the two replicates and is comparable to what the TAP-seq authors reported in Figure 2c of their paper. We added this statement in the manuscript: “This capture efficiency is comparable to the capture efficiency reported for TAP-seq (Schraivogel et al., 2023).”

There is a typo in 2A “CRSPRa” should be “CRISPRa”. There is also a typo on line 213.

Thank you for pointing these out, we have corrected them.

Regarding 2H, it would be helpful to annotate the mean expression level here. Similarly 2E I think could include all targeted TSSes and plot as a function of expression level. Clearly the expression level will affect which links can reliably be established.

Thank you for the suggestion. We have included additional panels to Figure 2 in which the gene expression of all genes in 2H is depicted as well as all targeted TSSes as a function of expression level in Figure S2C (shown here in Figure R8). We would like to point out that for each CaRE we have an average of 68 cells or experimental replicates, so we have the statistical power to reliably define pairs. Additionally, with the CRISPRa, as opposed to CRISPRi, we observe a gain of expression that can much more reliably be called than the loss of signal in a noisy whole-transcriptome CRISPRi screen.

A)

B)

Figure R8. A) TESLA-seq normalized gene expression level, comparing cells having gRNA targeting indicated gene promoters versus all other cells. B) Expression level of all differentially expressed target genes determined by TESLA-seq in perturbed and control cells.

When you talk about skipping genes, I assume you are counting noncoding transcripts as genes? Perhaps worth breaking this out as coding vs. non-coding. Are the skipped genes also just very low expression? That could explain why there is no obvious effect.

The skipped gene analysis considers only the 59 genes captured in our data, as shown in Figure 4A. We have annotated these genes with their expression levels and classified them as either coding or non-coding/pseudogenes. As summarized in the Venn diagram below, both coding and non-coding/pseudogenes include a representative proportion of lowly and highly expressed genes, but coding genes show an overall higher expression, whereas noncoding genes are mostly lowly expressed. We also performed the analysis separately for coding and non-coding genes and found that the results remained consistent with the analysis of all genes combined. In summary, we observe that the 59 captured genes have a roughly equal proportion of lowly and highly expressed genes, and there is no evidence that skipped genes are enriched in lowly expressed genes (Figure R9). We included this sentence in the results section: “In 71% (65) of significant cases, there is at least one gene between a CaRE and its target (Figure 3B), with an average of 7.7 skipped genes. These results are consistent also when the analysis is done separately on coding and non-coding genes (data not shown).”

A)

B)

C)

Figure R9. A) Venn diagram depicting the overlap of genes detected that are coding or noncoding and highly or lowly expressed. B) Relationship between adjusted p-values ($-\log_{10}$) of each CaRE-gene regulatory pair and number of genes located between the CaRE and the gene (jumped genes). Color

depicts the average fold change for coding (left) and noncoding genes (right). C) Histogram showing the number of genes located between a CaRE and a gene in the genome (jumped genes) for each CaRE-gene regulatory pair for coding (left) and noncoding genes (right).

Regarding 3G, training a random forest classifier on 100 examples will probably result in strong overfitting. Why not just use logistic regression or even simply compare summary statistics for the different features, given that you are preselecting the strongest examples in each class?

Both (regularized) logistic regression and random forests typically avoid overfitting, albeit by different strategies. Performing logistic regression to identify the most predictive features, results are very similar to the results of the random forest model. When combined, the most robust and consistent top features are accessibility and the presence of H3K4me1 mark at the CaREs. We have also included the summary statistics for all of the features included in the models in the supplemental Figure 3H-I and here in Figure R10. We added the following text in the manuscript: “Results are very similar when we perform logistic regression or compare individual features between significant and nonsignificant CaREs (Figure S3H-I).”.

A)

B)

Figure R10. Characteristics of TESLA-seq identified CaREs. A) Comparison of ATAC signal (left), H3K27ac signal (middle) and H3K4me1 signal (right) between significant and nonsignificant CaREs. B) CaRE features that contribute the most to the predictability of a logistic regression model.

In 4A, are all of the depicted genes reliably captured? I'm wondering again if expression level partly explains why some appear to have no regulators.

All 59 genes shown in Figure 4A are, by definition, "captured," meaning that after normalization, their total normalized expression across all cells exceeds one molecule/cell. The expression level box plot provides an overview of the variation in expression levels among these captured genes (Figure R11). We have included in the manuscript the expression level of all differentially expressed genes as part of Figure 2H as well as of all of the captured genes as a Supplementary Figure 5.

Figure R11. Expression level of all genes detected by TESLA-seq in perturbed and control cells.

There is an overall question of what is going on with the other 90% of fitness-defined CaREs, as only 60 CaREs participated in TESLA-seq interactions out of >600 identified in the original screen. Though the paper favors an interpretation of conservative selection, I do think two other alternatives are: (1) off-target effects of CRISPRa and (2) false positives from an overly permissive pipeline for the fitness data. I think some consideration should be given to these possibilities.

Thank you for the suggestions. In our statistical analysis, we used a stringent cutoff ($FDR < 0.05$) throughout the paper. Importantly, we assayed the top 222 CaREs (not >600) from the viability screen, since this is a number for which we could design one run of the targeted single-cell RNA-seq with sufficient numbers of replicates. These same CaREs were also selected as top hits with another commonly used tool - MAGECK. Of those 222 CaREs, 60 affected one or more target genes within the 6MB genomic space. We have clarified this further in the methods section.

In addition to conservative selection with stringent filtering criteria, several other possibilities remain as to why we did not identify additional targets within this space: 1) the gene is not expressed and is not activated by any of the gRNAs that we used in the study, the majority of the expressed genes we do capture (Figure 2C). 2) the target gene is within the 6MB genomic window that we analyzed but we were unable to assay it for various reasons, e.g. limitations of gene annotations such as poorly annotated (long) non-coding transcripts as we only captured genes for which a 3' end is well annotated and non-repetitive. 3) they could be affecting genes outside of the 6MB genomic space as part of their direct mechanism of action or could be acting post-transcriptionally (e.g. a long non-coding RNA affecting translation). 4) they could be false positives or off-target effects. Thus, many factors may contribute to prevent a detection for all of them; in turn, we can be confident of the ones for which we detect the target gene: We quantify a change in transcription replicated in an average of 68 cells, compared to e.g. the loss of signal in a noisy scRNA-seq experiment as in perturb-seq like screens.

A final point worth considering is whether the tiling fitness screen will ultimately be necessary. Various compressed experimental designs for studying enhancers have been put forward, including the Gasperini et al. paper. It seems possible that these could be used alongside the single-cell readout here to do tiling directly.

Indeed, one could completely bypass the viability screen. However, the effect of a regulatory element on a cellular phenotype is useful information in addition to the transcriptome. Additionally, it helps to pinpoint functional gRNAs and elements and reduce the number of gRNAs that are being assayed in a costly scRNA-seq experiment. With the cost of sequencing and single-cell RNA-seq experiments dropping continuously, it should be eventually possible to assay all ~50k gRNAs in a single-cell RNA-seq experiment within a reasonable budget. This opens up the possibility for genome-wide screens in multiple cell lines and with multiple different CRISPR constructs to functionally dissect the human genome.

We commented on this in the discussion: “TESLA-seq is a stand-alone approach and can be used to directly redout perturbation effects without any phenotypic preselection as described in this manuscript.”

References:

1. Damoczi, J., Knoops, A., Martou, M.-S., Jaumaux, F., Gabant, P., Mahillon, J., Veening, J.-W., Mignolet, J., and Hols, P. (2024). Uncovering the arsenal of class II bacteriocins in salivarius streptococci. *Commun. Biol.* 7, 1511. <https://doi.org/10.1038/s42003-024-07217-y>.
2. Santos-Júnior, C.D., Torres, M.D.T., Duan, Y., Rodríguez del Río, Á., Schmidt, T.S.B., Chong, H., Fullam, A., Kuhn, M., Zhu, C., Houseman, A., et al. (2024). Discovery of antimicrobial peptides in the global microbiome with machine learning. *Cell* 187, 3761-3778.e16. <https://doi.org/10.1016/j.cell.2024.05.013>.
3. Coelho, L.P., Alves, R., del Río, Á.R., Myers, P.N., Cantalapiedra, C.P., Giner-Lamia, J., Schmidt, T.S., Mende, D.R., Orakov, A., Letunic, I., et al. (2022). Towards the biogeography of prokaryotic genes. *Nature* 601, 252–256. <https://doi.org/10.1038/s41586-021-04233-4>.
4. Hsieh, T.C., Ma, K.H., and Chao, A. (2016). iNEXT: an R package for rarefaction and extrapolation of species diversity (Hill numbers). *Methods Ecol. Evol.* 7, 1451–1456. <https://doi.org/10.1111/2041-210X.12613>.
5. Gavriilidou, A., Kautsar, S.A., Zaburannyi, N., Krug, D., Müller, R., Medema, M.H., and Ziemert, N. (2022). Compendium of specialized metabolite biosynthetic diversity encoded in bacterial genomes. *Nat. Microbiol.* 7, 726–735. <https://doi.org/10.1038/s41564-022-01110-2>.
6. Gao, Y., Zhong, Z., Zhang, D., Zhang, J., and Li, Y.X. (2024). Exploring the roles of ribosomal peptides in prokaryote-phage interactions through deep learning-enabled metagenome mining. *Microbiome* 12. <https://doi.org/10.1186/s40168-024-01807-y>.
7. Sugrue, I., Ross, R.P., and Hill, C. (2024). Bacteriocin diversity, function, discovery and application as antimicrobials. *Nat. Rev. Microbiol.* <https://doi.org/10.1038/s41579-024-01045-x>.
8. Ma, Y., Guo, Z., Xia, B., Zhang, Y., Liu, X., Yu, Y., Tang, N., Tong, X., Wang, M., Ye, X., et al. (2022). Identification of antimicrobial peptides from the human gut microbiome using deep learning. *Nat. Biotechnol.* 40, 921–931. <https://doi.org/10.1038/s41587-022-01226-0>.
9. King, A.M., Zhang, Z., Glassey, E., Siuti, P., Clardy, J., and Voigt, C.A. (2023). Systematic mining of the human microbiome identifies antimicrobial peptides with diverse activity spectra. *Nat. Microbiol.* 8, 2420–2434. <https://doi.org/10.1038/s41564-023-01524-6>.
10. Chen, J., Jia, Y., Sun, Y., Liu, K., Zhou, C., Liu, C., Li, D., Liu, G., Zhang, C., Yang, T., et al. (2024). Global marine microbial diversity and its potential in bioprospecting. *Nature* 633, 371–379. <https://doi.org/10.1038/s41586-024-07891-2>.

Referees' reports, second round of review

Reviewer #1:

The authors have satisfactorily resolved all my comments.

Reviewer #2:

I am mostly happy with the authors responses to my questions. I would potentially temper the claims about dCas9-KRAB not working well, as I think this is likely more a cell line issue than anything else. I remain somewhat concerned that there are false positives driven by off-target binding, which the current experiments cannot fully exclude, but I agree that many of the elements are clearly functional. I also think the concept of CaREs is both important and interesting, and the experimental methodology here is novel and sound. I am supportive of publication.

Authors' response to the second round of review

Reviewer #1: The authors have satisfactorily resolved all my comments.

Thank you very much for insightful comments.

Reviewer #2: I am mostly happy with the authors responses to my questions. I would potentially temper the claims about dCas9-KRAB not working well, as I think this is likely more a cell line issue than anything else. I remain somewhat concerned that there are false positives driven by offtarget binding, which the current experiments cannot fully exclude, but I agree that many of the elements are clearly functional. I also think the concept of CaREs is both important and interesting, and the experimental methodology here is novel and sound. I am supportive of publication.

Thank you very much for insightful comments. We have added the following sentence to the Supplementary Note 1: "Of note, since different CRISPRi constructs can have different efficacies in different cell lines, it is also possible that dCas9-KRAB is not an efficient repressor in the SH-SY5Y cell line that we used in the study."